# The opposing forces of shear flow and sphingosine-1-phosphate control marginal zone B cell shuttling

Kerry Tedford[1], Michael Steiner[1], Stanislav Koshutin[1], Karin Richter[1], Laura Tech[1,2], Yannik Eggers[1], Inga Jansing[1], Kerstin Schilling [1], Anja Erika Hauser[2], Mark Korthals[1] & Klaus-Dieter Fischer [1]

Splenic marginal zone B cells (MZB) shuttle between the blood-filled marginal zone for antigen collection and the follicle for antigen delivery. However, it is unclear how MZBs migrate directionally from the marginal zone to the follicle. Here, we show that murine MZBs migrate up shear flow via the LFA-1 (αLβ2) integrin ligand ICAM-1, but adhere or migrate down the flow via the VLA-4 integrin (α4β1) ligand VCAM-1. MZBs lacking *Arhgef6* (Pak-interacting exchange factor (αPIX)) or functional LFA-1 are impaired in shuttling due to mislocalization toward the VCAM-1-rich red pulp. Sphingosine-1-phosphate (S1P) signaling through the S1PR3 receptor inhibits MZB migration up the flow, and deletion of *S1pr3* in *Arhgef6*$^{-/-}$ mice rescues mislocalized MZBs. These findings establish shear flow as a directional cue for MZB migration to the follicle, and define S1PR3 and VCAM-1 as counteracting forces that inhibit this migration.

[1] Institute for Biochemistry and Cell Biology; Health Campus Immunology, Infectiology and Inflammation, Otto-von-Guericke University Magdeburg, 39120 Magdeburg, Germany. [2] Deutsches Rheuma-Forschungszentrum Berlin and Immunodynamics, Charite Universitätsmedizin, Charitéplatz 1, 10117 Berlin, Germany. Kerry Tedford, Michael Steiner, and Stanislav Koshutin contributed equally to this work. Correspondence and requests for materials should be addressed to K.T. (email: kerry.tedford@med.ovgu.de) or to K.-D.F. (email: klaus.fischer@med.ovgu.de)

Murine marginal zone B cells (MZB) are a subset of B cells that sieve antigens circulating in the blood[1,2]. Blood percolates out of openings in the marginal sinus that borders the B-cell follicles in the spleen and that marks the inner edge of the marginal zone[3]. MZBs sample the blood for antigens as it pools in the marginal zone before flowing into the red pulp, to be collected in venous sinuses, and returned to the circulation. Thus, MZBs are exposed to a directional flow of blood, with a vector that originates from the marginal sinus at follicles and that orients toward the red pulp. Directional flows in blood vessels exert shear force that activates integrins, such as very late antigen 4 (VLA-4) (α4β1) and lymphocyte function-associated antigen 1 (LFA-1) (αLβ2), to bind to their ligands, vascular cell adhesion protein 1 (VCAM-1), and intercellular adhesion molecule 1 (ICAM-1), enabling T cells to adhere or migrate against the force of the flow[4,5]. However, it is unknown if MZBs exposed to the shear forces of blood adhere to the marginal zone, or migrate in a specific manner.

MZBs were formerly described as sessile because these cells do not recirculate between lymphoid organs or in blood, and were found only in the marginal zone in rodents. However, Cyster and colleagues showed that MZBs shuttle constitutively between the marginal zone and the interior of follicles to deliver antigens[6–8]. MZB shuttling in and out of follicles is regulated by CXCL13 and by sphingosine-1-phosphate (S1P)[7,8]. The CXCR5/CXCL13 signaling axis is required for MZB entry into follicles, whereas S1P signaling is required for MZB exit from the follicles[7,8]. S1P is a signaling phospholipid that is produced by red blood cells, and is present at high concentrations in the blood but at low concentrations inside follicles and in the red pulp[9,10]. Among the five S1P receptors, MZBs primarily express S1PR1 and S1PR3. S1PR1 is expressed at high levels on MZBs, as well as many other immune cell types, while S1PR3 is expressed at lower levels than S1PR1 on MZBs but is enriched on MZBs[11]. S1PR1 is internalized upon S1P signaling, and this event is also a requirement for MZB entry into follicles[12,13]. S1PR3 is essential for the normal assembly of endothelial cells that line the marginal sinus and, consequently, the organization of the marginal zone[14]. However, the reason for the MZB-specific expression of S1PR3 is unknown. Additionally, how MZBs migrate in the direction of the follicle from the marginal zone is also unknown. Although CXCR5 signaling is required for follicular entry, it is unclear if a soluble or haptotactic chemokine gradient could be established in the flow from the sinus to the outer border of the marginal zone.

LFA-1 and VLA-4 integrins are key to migration of lymphocytes but are not interchangeable. Studies have suggested that LFA-1 evolved in tandem with adaptive immunity to perform functions specific to T and B lymphocytes[15]. In T cells migrating under shear flow, LFA-1 is distributed around the entire contact area, while VLA-4 is localized at the rear[16]. Moreover, LFA-1 dominates over VLA-4 in supporting shear-resistant T-cell crawling and transendothelial migration[16]. Additionally, T cells migrating under shear flow move directionally up the flow when adhering to ICAM-1 but move down the flow when bound to VCAM-1[17,18]. One structural reason for these differences is that the alpha subunit of LFA-1 (αL) possesses an I-domain, an additional loop that the alpha subunit of VLA-4 lacks. The I-domain hinders LFA-1 ligand binding relative to VLA-4 ligand binding and requires an additional activation signal[15], suggesting that LFA-1 functions in migration are distinct from those of VLA-4.

We investigated whether MZBs respond to shear flow with either adhesion or migration and if S1PR3 is involved. Here we show that shear flow strongly activates MZB migration up the flow via LFA-1 adherence to ICAM-1, and that VLA-4 adherence to VCAM-1 or S1P signaling via S1PR3 inhibits this directional migration. MZBs with a mutation in *Arhgef6* are faster and less adherent than wild-type MZBs. *Arhgef6*−/− MZBs are partly mislocalized to the red pulp just outside the marginal zone; however, MZB positioning is normal in *Arhgef6*−/− *S1pr3*−/- double-knockout mice. These findings identify shear flow as a force activating the directional migration of MZBs to the follicle, and show that this directional migration is restrained by VCAM-1 and S1P.

## Results

**MZBs migrate up the shear flow**. Marginal zone B cells (MZBs) adhere more than follicular B cells (FOB) to intercellular adhesion molecule 1 (ICAM-1) and vascular cell adhesion protein 1 (VCAM-1) under static conditions[19]. Thus, we wanted to determine whether this increased adhesion by MZBs also enables these cells to adhere under shear flow. FOBs or MZBs were incubated in flow chambers coated with ICAM-1, VCAM-1, or a mixture of both. The cells were counted after the application of flow to determine the rate of detachment at 3 dyn cm−2. FOBs detached rapidly over the 30 min of imaging, and final counts ranged from 47 to 62% of starting numbers, whereas 86 to 94% of MZBs remained at 30 min, demonstrating that shear flow could easily induce the detachment of FOBs but not MZBs (Fig. 1a).

To test the response of MZBs adhering to ICAM-1 to increasing levels of flow strength, we assessed several parameters of migratory behavior over 30 min: the migration index, velocity, and straightness of the migration track. In the absence of shear flow (0 dyn cm−2), MZBs spontaneously migrated ~75 μm in all directions. The lack of directionality was confirmed by the approximately equal numbers of tracks that terminated above (black tracks) and below (red tracks) the x axis (Fig. 1b). Also, the migration index was centered at 0, indicating no net gain in directional movement (Fig. 1c). At a relatively low 1 dyn cm−2 flow, MZBs again migrated equally in all directions, but the average distance was increased ~33% (Fig. 1b, c). However, at a flow strength of 3 dyn cm−2, a majority of the MZB tracks appeared as black, indicating that the MZBs detected the flow and responded by migrating up the flow (Fig. 1b). At flow strengths of 6 and 10 dyn cm−2, the velocity remained constant, while the migration index and straightness increased (Fig. 1c). The increased migration of MZBs was not due to differences in viability between MZBs and FOBs (Supplementary Figs. 1a and 2), and was specific to ICAM-1 coating on the slides with blocking by bovine serum albumin (Supplementary Fig. 1b, c). These data showed that MZBs sense shear flow and re-orient their migration polarity complexes in the direction of the flow, with the efficiency of this process increasing with the strength of the flow.

**MZB migration on MAdCAM-1 and CXCL13**. To investigate how ligand composition affects MZBs, we tested MZB migration at a flow of 8 dyn cm−2 on combinations of ICAM-1 and VCAM-1 totaling 5 μg ml−1, from ICAM-1 alone to VCAM-1 alone. The track plots showed that MZBs migrated well up the flow on ICAM-1 alone but were inhibited from migration in the absence of ICAM-1, when only VCAM-1 was present (Fig. 2a). The inclusion of VCAM-1 did not affect the migration index when it was half the mixture or less. When the proportion of VCAM-1 in the mix was increased to 75–100%, the velocity, migration up the flow, and straightness parameters decreased by half (Fig. 2b). Additionally, the proportion of MZBs that migrated >10 μm away from the start position (displacement >10) decreased from ~80% on ICAM-1 alone to ~35% on VCAM-1 alone (Fig. 2b). To determine the in vivo relevance of MZB responses to ICAM-1 and VCAM-1, we stained for these integrin ligands in splenic follicles and observed that ICAM-1 was strongly expressed in the

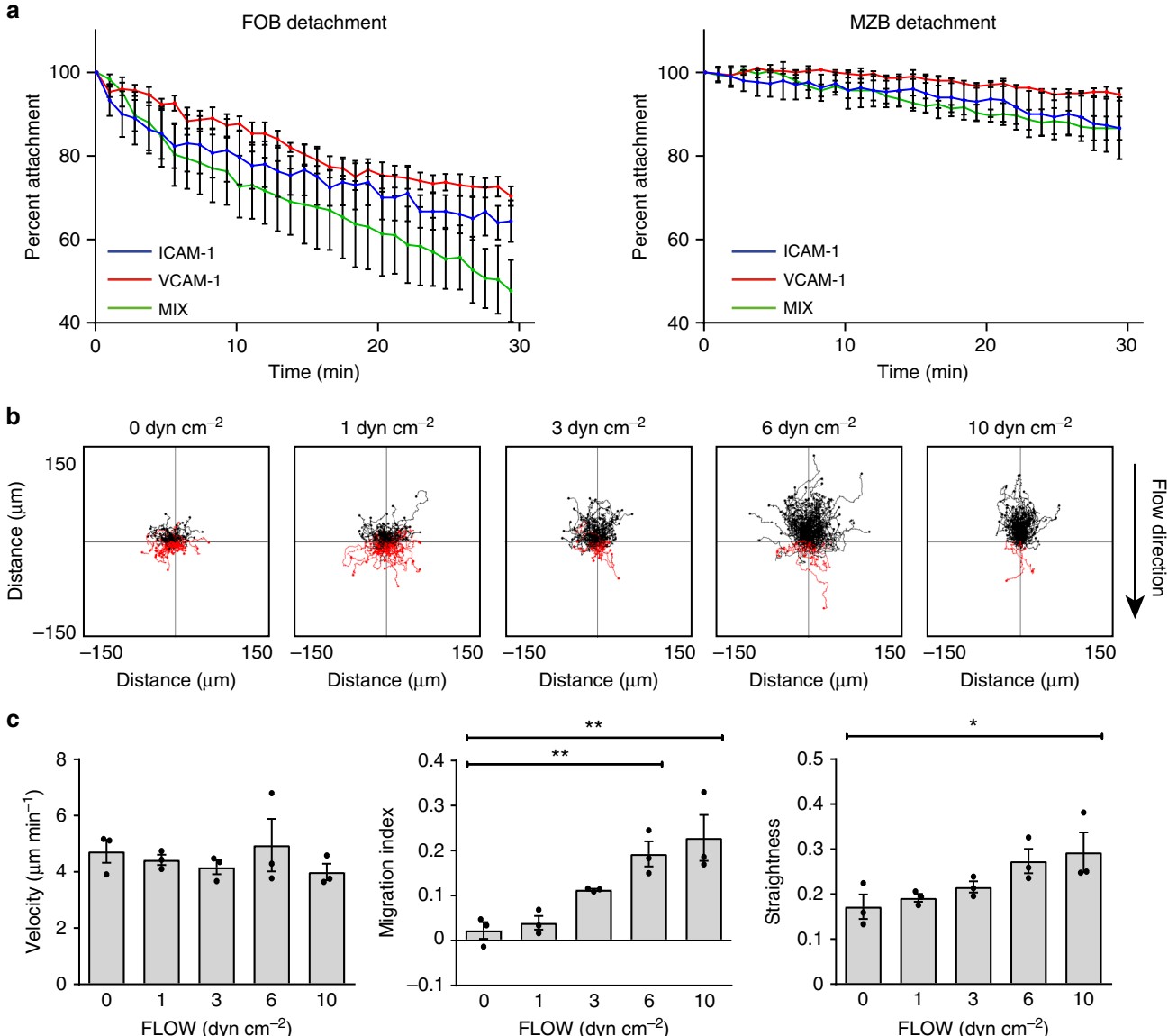

**Fig. 1** MZB are not flushed out by shear flow but migrate up it. **a** Detachment over time of marginal zone B cells (MZB) and follicular B cells (FOB). Percent attachment based on number of cells after flow (3 dyn cm$^{-2}$) start ($t = 10$ s), mean ± SEM on ICAM-1 (2 µg ml$^{-1}$), VCAM-1 (2 µg ml$^{-1}$), or ICAM-1 and VCAM-1 (1 µg ml$^{-1}$ each) mixed ($n = 3$). **b** Representative track plots of MZB migration up the flow on ICAM-1 (7 µg ml$^{-1}$). Flow strengths indicated in dyn cm$^{-2}$ by the number over the track plot; flow direction indicated by arrow. Tracks with end points above the horizontal axis are shown in black; tracks below are shown in red. **c** Quantification of velocity, migration index, and straightness for the indicated flow strengths. All three graphs show different parameters from the same set of experiments; bars show mean ± SEM. Data are from three experiments with 1 mouse each. Symbols in one condition group denote the three replicates and represent the average of 100–200 cells from each individual mouse. *$P < 0.05$, **$P < 0.01$, one-way ANOVA with Dunnett post hoc tests comparing all flow strengths to 0 dyn cm$^{-2}$

marginal zone while VCAM-1 was primarily found in the red pulp, consistent with the findings of Lu et al[19]. Despite high expression in the red pulp, VCAM-1 expression was relatively low in the marginal zone (Fig. 2c). At higher exposure levels of the image, some faint VCAM-1 signals were detectable but at correspondingly lower levels than the ICAM-1 signals. To investigate whether a reduced proportion of VCAM-1 affected MZB migration, we examined migration using ICAM-1 to VCAM-1 ratios that began with a 10:1 ratio of ICAM-1 to VCAM-1; however, we did not detect any effects of VCAM-1 at this low level and only observed migration inhibition by VCAM-1 when present in over a 1:1 ratio with ICAM-1 (Supplementary Fig. 3). Collectively, our in vivo migration assays and the distribution of ICAM-1 in the marginal zone and VCAM-1 in the red pulp,

suggest that MZBs migrate up the flow in the marginal zone but are restrained in the red pulp.

We next tested MZB migration up the flow on ligands or chemokines that affect MZB shuttling or are present in the splenic follicle system. CXCL13 is essential for MZB entry into follicles[7]. While it is not known whether CXCL13 functions in a soluble or immobilized form, it seems unlikely that any type of soluble gradient could be established in an area with constant flow. Therefore, we coated slides with ICAM-1 and either CXCL13 or CXCL12, another chemokine with no major effects on MZB migration. We found no obvious CXCL12-dependent effects; however, CXCL13 clearly inhibited the migration of MZBs. Although their directional migration was reduced, their overall ability to move, as shown by the % of MZBs displaced by

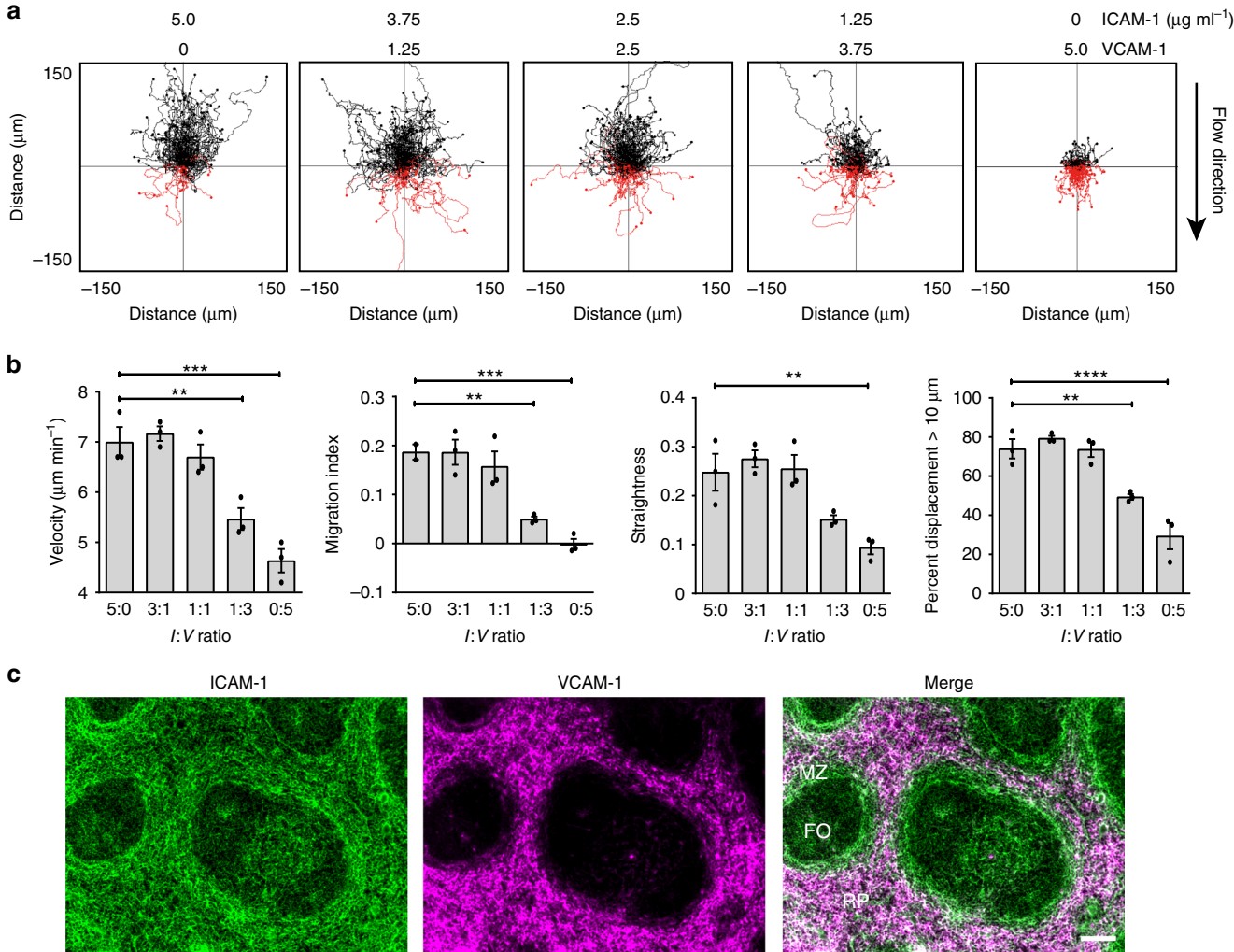

**Fig. 2** MZBs migrate up the flow on ICAM-1 but not on VCAM-1. **a** Representative track plots of MZB migration up the flow (8 dyn cm$^{-2}$; direction indicated by arrow) with coating (ICAM-1 or VCAM-1) amounts shown above. **b** Quantification of velocity, straightness, migration index, and % of cells with displacement >10 μm for the ICAM-1 to VCAM-1 (IV) ratios shown above in **a**. All four graphs show different parameters from the same set of experiments; bars show mean ± SEM. Data are from three experiments with one mouse each. Symbols in one condition group denote the three replicates and represent the average of 100–200 cells from each individual mouse. **c** Microscopy of ICAM-1 (green), VCAM-1 (magenta), and the merge (white) in the spleen, showing follicles (FO), marginal zone (MZ), and red pulp (RP). Data represent two experiments with one spleen each. Scale bar; 100 μm. **P < 0.01, ***P < 0.001, ****P < 0.0001, one-way ANOVA with Dunnett post hoc tests comparing all ICAM-1/VCAM-1 mixtures to ICAM-1 alone (5:0)

>10 μm, was also reduced, suggesting that CXCL13 enhances MZB adhesion (Fig. 3a, b). We next tested several other integrin ligands that could potentially affect MZB migration, including ICAM-2 and mucosal vascular addressin cell adhesion molecule 1 (MAdCAM-1). ICAM-2 did not negatively affect MZB migration and worked as well as ICAM-1 (Fig. 3c, d). However, MAdCAM-1 is expressed on endothelial cells lining the marginal sinus, and MZBs adhered so strongly to MAdCAM-1 that they could not migrate at all (Fig. 3c, d). These data show that CXCL13 and MAdCAM-1, two ligands co-expressed on the sinus lining the follicle[20], cause MZBs to adhere strongly and reduce their migration up the flow.

**LFA-1 and blood flow affect MZB positioning in vivo.** Splenic follicles are formed around the terminal ends of arterioles that release blood into a sinus perforated with openings into the marginal zone. To test whether blood flow influences MZB migration in vivo, and by extension, MZB positioning, we blocked lymphocyte function-associated antigen 1 (LFA-1) integrin

functioning for 1 h to induce MZBs to dislodge from the marginal zone. Although both LFA-1 and very late antigen 4 (VLA-4) must be blocked together to cause a complete release of MZBs out of the spleen and into the blood in 3–6 h[19,21], we reasoned that if ICAM-1 is the main ligand in the marginal zone, then blocking LFA-1 must affect MZBs. We used 5-min injections of αCD21 to mark the boundaries of the marginal zone[8], and we co-stained spleen sections with IgM to determine whether any MZBs were forced outside of the marginal zone by outward flow and decreased adhesion after LFA-1 blockade for 1 h. Indeed, we observed an increase in IgM-positive cells in areas outside the CD21 staining in the marginal zone that was not present in control-injected mice (Fig. 4a, Supplementary Fig. 4a, Supplementary Note 1). To confirm that CD21 labeling was specific for the marginal zone, we injected αF4/80, a marker of red pulp macrophages, at the same time as αCD21 and observed that this marker did not stain the marginal zone but moved through to the red pulp (Supplementary Fig. 4b). Conversely, the number of MZBs shuttling into the follicle was reduced in mice treated with αLFA-1 for 1 h (Fig 4b) and 2 h (Supplementary Fig. 5), as shown

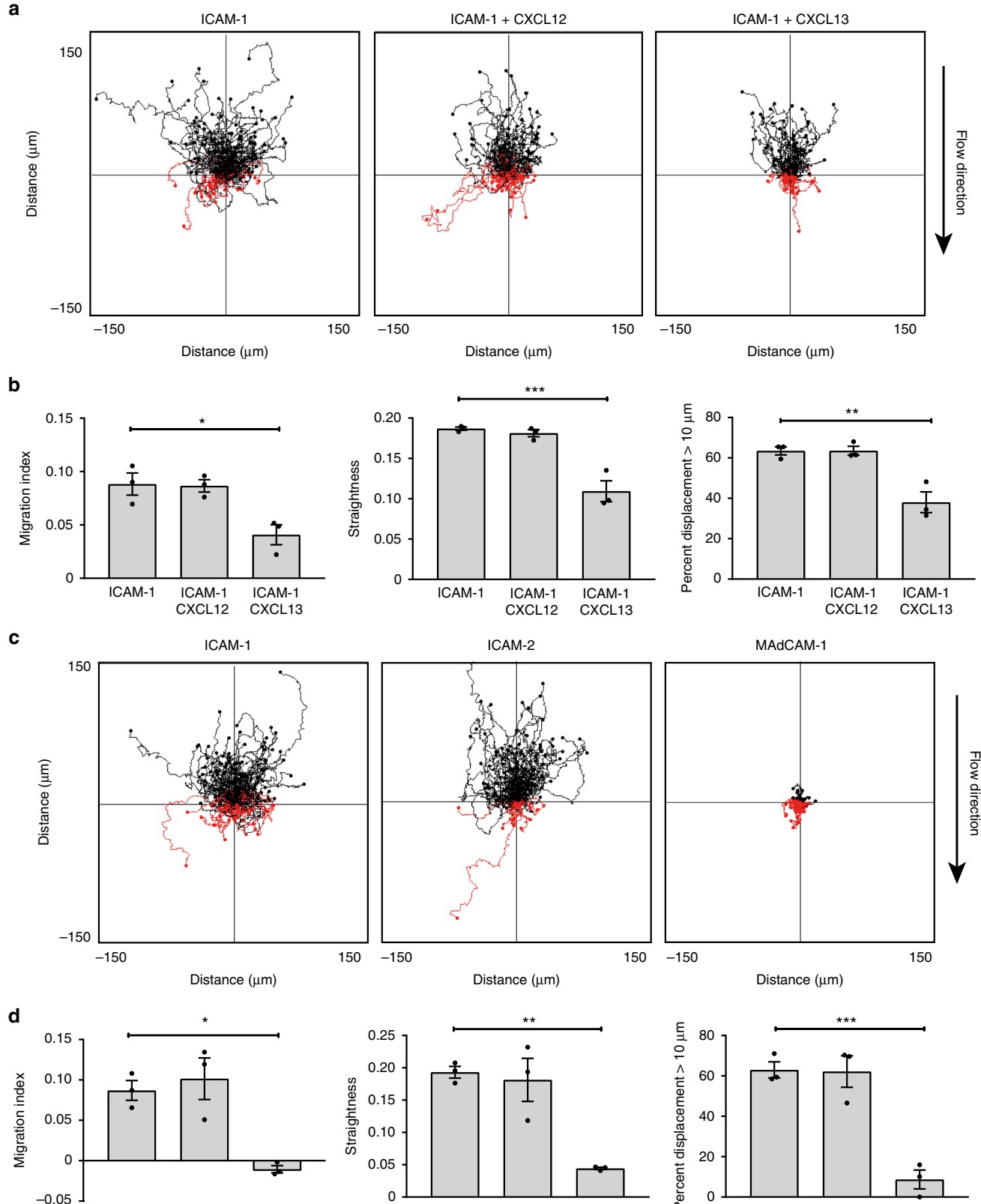

**Fig. 3** CXCL13 and MAdCAM-1 inhibit MZB migration up the flow. **a** Representative track plots for MZB migrating up flow (3 dyn cm$^{-2}$) on ICAM-1 (5 µg ml$^{-1}$) alone or in combination with coated CXCL12 (0.3 µg ml$^{-1}$) or with CXCL13 (2 µg ml$^{-1}$). **b** Quantification of migration index, straightness, and % of cells that displaced more than 10 µm. All three graphs show different parameters from the same set of experiments; bars show mean ± SEM. Data are from three experiments with one mouse each. **c** Representative track plots for MZB migrating up the flow (3 dyn cm$^{-2}$) on ICAM-1 (5 µg ml$^{-1}$), ICAM-2 (5 µg ml$^{-1}$), or MAdCAM-1 (5 µg ml$^{-1}$). **d** Quantification as in **b**. $*P < 0.05$, $**P < 0.01$, $***P < 0.001$, one-way ANOVA with Dunnett post hoc tests comparing all ligands/chemokines to ICAM-1 alone (5:0)

by a double in vivo labeling assay developed by Cinamon et al.[8] using αCD35 and αCD21 injected at intervals of 10 and 20 min, respectively. These results suggested that MZBs lacking adhesion to ICAM-1 were pushed away from the follicle by the force of blood flow toward the red pulp.

To directly test the effects of blood flow, we measured shuttling into the follicle in spleens in which the blood supply was turned off by excising the spleens. As previously shown[8], a 5-min injection of αCD21 did not reveal any shuttling cells inside the follicle, while a 30-min injection revealed the extent of shuttling

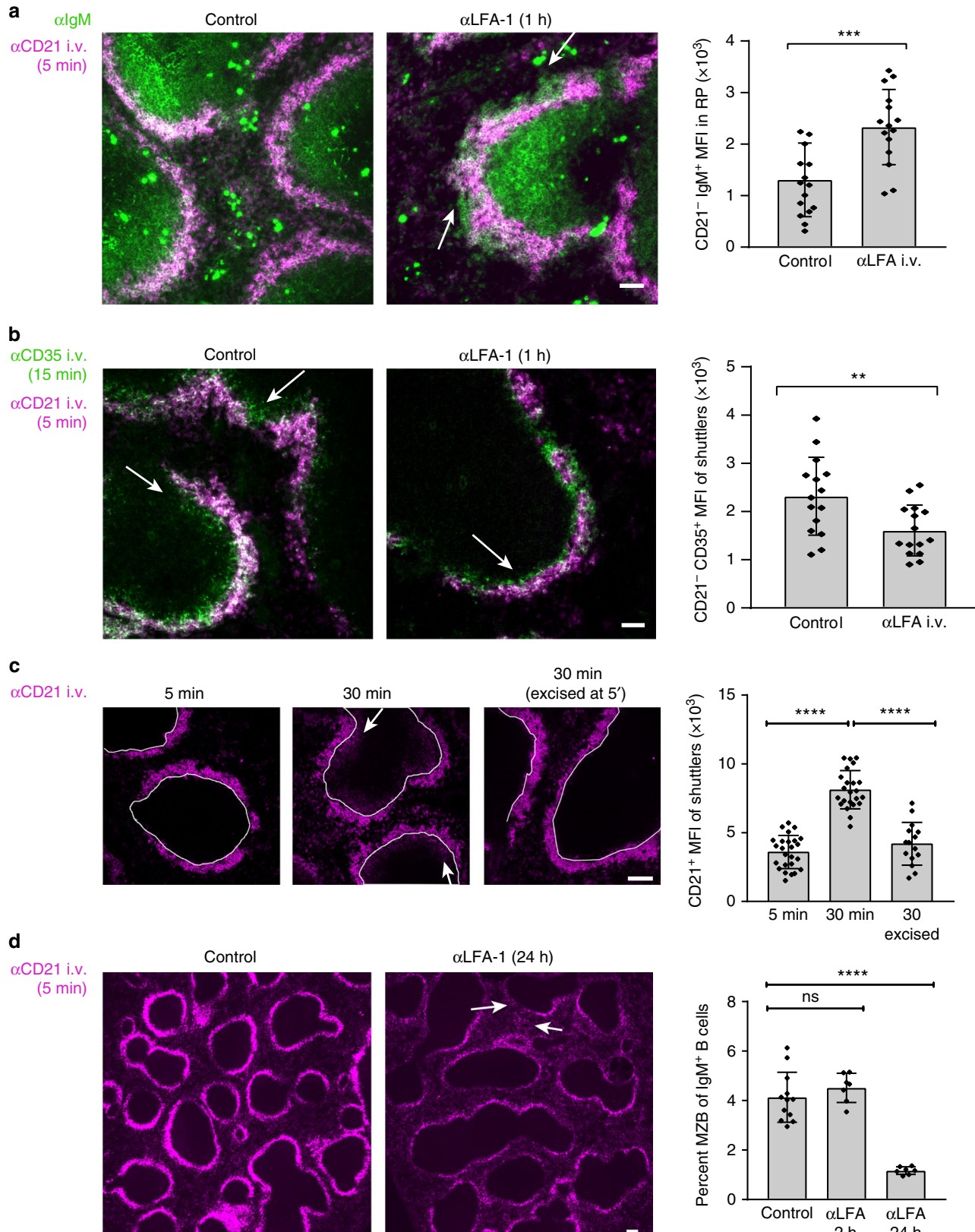

inside the follicle, past the MAdCAM-1[+]-lining sinus border (Fig. 4c). However, in spleens that had the blood flow cut off after 5 min, the amount of shuttling cells was reduced, consistent with flow as a force that induces MZB shuttling into the follicle.

If flow is indeed directed outward from the follicle, and if LFA-1 is the principal integrin that is active in the marginal zone, it is possible that blocking LFA-1 for sustained periods of time would result in a loss of MZBs from the marginal zone toward the red pulp and into circulation. We blocked LFA-1 with an injection of anti-LFA-1 antibody for 24 h and then labeled MZBs with a 5-min αCD21 injection. The overall number of MZBs in spleens decreased >65%, and the histology revealed that the marginal zones of treated mice were almost completely eroded (Fig. 4d). The relatively low amount of VCAM-1 in the marginal zone likely suffices to hold MZBs for a few hours with blocked LFA-1 but is not sufficient to enable MZBs to withstand the flow for 24 h, and the MZBs are lost from the spleen. The remaining MZBs were dispersed between the follicles, suggesting that the loss of LFA-1 adhesion combined with blood flow through the marginal zone causes the displacement of MZBs down the flow and out to the red pulp (Fig. 4d, white arrows). These experiments, along with our finding that MZBs migrate up the flow on ICAM-1 (Figs. 1, 2, and 3), support a dual function for LFA-1/ICAM-1 interactions: LFA-1 prevents MZBs from being washed out of the spleen by the force of blood flow and supports their migration up the flow into the follicle.

**S1P inhibits MZBs from migrating up the flow.** Sphingosine-1-phosphate (S1P) exerts many varied effects in multiple cell systems, often by controlling cell-cell and cell-matrix adhesion[22]. The S1P receptor S1PR1 is essential for shuttling MZBs to exit the follicle[7]. Therefore, we speculated that S1P affects MZB migration up the flow. We tested S1P treatment on the shear flow-induced migration of wild-type MZBs on ICAM-1. Track plots of MZBs treated with 200 or 600 nM S1P displayed more red tracks than untreated MZBs, and a slight reduction in black tracks extending upward was also observed (Fig. 5a). The velocity of MZBs migrating up the flow was unaffected by S1P, but the migration index and straightness parameters revealed that S1P reduced the ability of MZBs to migrate directionally (Fig. 5b). Additionally, there was a modest increase in MZBs that were displaced by >10 μm at 200 nM S1P, suggesting that S1P affects LFA-1 (Fig. 5b). However, the detachment of S1P-treated MZBs under flow was unchanged (Fig. 5c); thus, adhesion is not reduced by S1P. Together, these results show that MZB migration up the flow is inhibited by S1P, which likely acts on LFA-1 activation or positioning.

**Increased migration and detachment of Arhgef6[−/−] MZBs.** The effects of flow on MZBs with blocked LFA-1 integrin prompted

us to test whether MZBs with a different type of adhesion defect would also be mislocalized toward the red pulp by the force of flow. To this end, we used lymphocytes with a mutation in Arhgef6. T cells and B cells lacking this cytoskeleton regulator have reduced cell–cell contacts and migrate faster than wild-type cells in transwell assays and in thymic lobes, consistent with a defect in adhesion[23,24]. We first quantified the numbers of Arhgef6[−/−] MZBs by flow cytometry and found an ~50% increase in their numbers in mice of all ages (Supplementary Fig. 6a). We next assessed Arhgef6[−/−] MZBs in transwell migration assays and observed that these cells migrate through the membrane in substantially greater numbers than wild-type MZBs under every condition tested, despite having normal surface levels of LFA-1 and VLA-4 integrins (Supplementary Fig. 7a, b). We tested Arhgef6[−/−] MZB migration under shear flow on ICAM-1 and VCAM-1 and found that Arhgef6[−/−] MZBs migrate ~50% faster than wild-type MZBs on ICAM-1 but not VCAM-1, resulting in longer, straighter, and more directional tracks on ICAM-1 (Fig. 6a, b). At higher amounts of VCAM-1, both wild-type and Arhgef6[−/−] MZBs migrated down the flow but Arhgef6[−/−] MZBs had increased velocity and migration index (Supplementary Fig. 8a, b). Given that cell speed is a function of adhesion[25–27], we tested Arhgef6[−/−] MZB adhesion under static and flow conditions. There were no differences between wild-type and Arhgef6[−/−] MZB adhesion to ICAM-1 under static conditions (Fig. 6d). In contrast, a flow adhesion assay designed to measure the detachment of MZBs under increasing flow strengths (from 3 to 15 dyn cm[−2]) showed that Arhgef6[−/−] MZBs detached more on ICAM-1 alone or an ICAM-1/VCAM-1 mix compared to wild type (Fig. 6c). These findings demonstrate that compared with wild-type cells, Arhgef6[−/−] MZBs migrate faster up the flow on ICAM-1 and down the flow on VCAM-1, and detach more on ICAM-1 under flow.

We also tested whether S1P inhibited the migration of Arhgef6[−/−] MZBs on ICAM-1 and VCAM-1. As shown above in Fig. 5, wild-type MZB migration up the flow was constrained, as revealed by the migration tracks (Fig. 6a) and by the moderate reduction in the wild-type migration index ($P = 0.09$) (Fig. 6a, b). However, the effects of S1P treatment on Arhgef6[−/−] MZBs were even stronger: the track plots showed a clear reversal in migration direction, from up the flow to down the flow, and there was a strong decrease in the migration index (Fig. 6b). In addition, as shown in Fig. 5, S1P induced a modest increase ($P = 0.09$) in the number of MZBs on ICAM-1 that migrated >10 μm from their starting point (Fig. 6b). The counter-flow effect of S1P was detectable for Arhgef6[−/−] MZBs on both ICAM-1 and VCAM-1. Thus, while the results of the S1P treatment of wild-type MZBs were clear, the effects were even more pronounced on Arhgef6[−/−] MZBs. Two-way analysis of variance (ANOVA) analysis of the results showed a significant interaction between S1P treatment

**Fig. 4** Influence of blood flow on positioning of MZB in vivo. **a** Immunofluorescence microscopy of IgM[+] cells located beyond the CD21-positive boundary of the MZ 1 h after i.v. injection of αLFA-1. Right panel: quantification of mean fluorescence intensity (MFI) in a band of 35 μm of CD21[−] IgM[+] cells outside the CD21 boundary of the MZ. Arrow = IgM[+] B cells outside the CD21-stained MZ. **b** Immunofluorescence microscopy of CD35[+] MZB shuttling into follicles 1 h after i.v. αLFA-1 injection. MZB that shuttled into the follicle were positive for αCD35 (i.v. 15 min) and negative for αCD21 (i.v. 5 min). Right panel: quantification of MFI inside a band of 70 μm inside of the marginal sinus. For **a**, **b**, symbols in one condition group denote individual follicles, 5 per condition in one experiment, from 3 mice each of control or αLFA-1 injected, 3 experiments total. Arrows = CD35[+] CD21[−] shuttling MZB inside the follicle. **c** Blood flow in spleen required for normal numbers of shuttling MZB. Confocal immunofluorescent images of follicles at 5 min after αCD21 injection ($n = 5$ mice), 30 min after injection ($n = 5$ mice), or 30 min after injection with the spleen excised at 5 min ($n = 3$ mice). MZB were identified by CD21 staining (white arrows) inside the follicle (MAdCAM-1 used to define the inner border, shown as a white line). Right panel: quantification of CD21[+] shuttling B cells inside the follicle, 5 follicles per mouse. **d** Immunofluorescence microscopy of CD21[+] MZB in the spleen following 24 h of i.v. αLFA-1 injection followed by 5 min of αCD21 i.v. injection. White arrows indicate thinned MZ and increased CD21 staining outside the MZ. Images representative of 4 experiments for control ($n = 12$) mice; 2 experiments each for αLFA-1 for 2 h ($n = 7$ mice) or 24 h ($n = 7$ mice). Data are expressed as the mean ± SEM. **$P < 0.01$, ***$P < 0.001$, ****$P < 0.0001$, by $t$ test (**a**, **b**) or one-way ANOVA (**c**, **d**) relative to middle column in **c** and first column in **d**. Scale bar, 50 μm for **a**, **b** and 100 μm for **c**, **d**

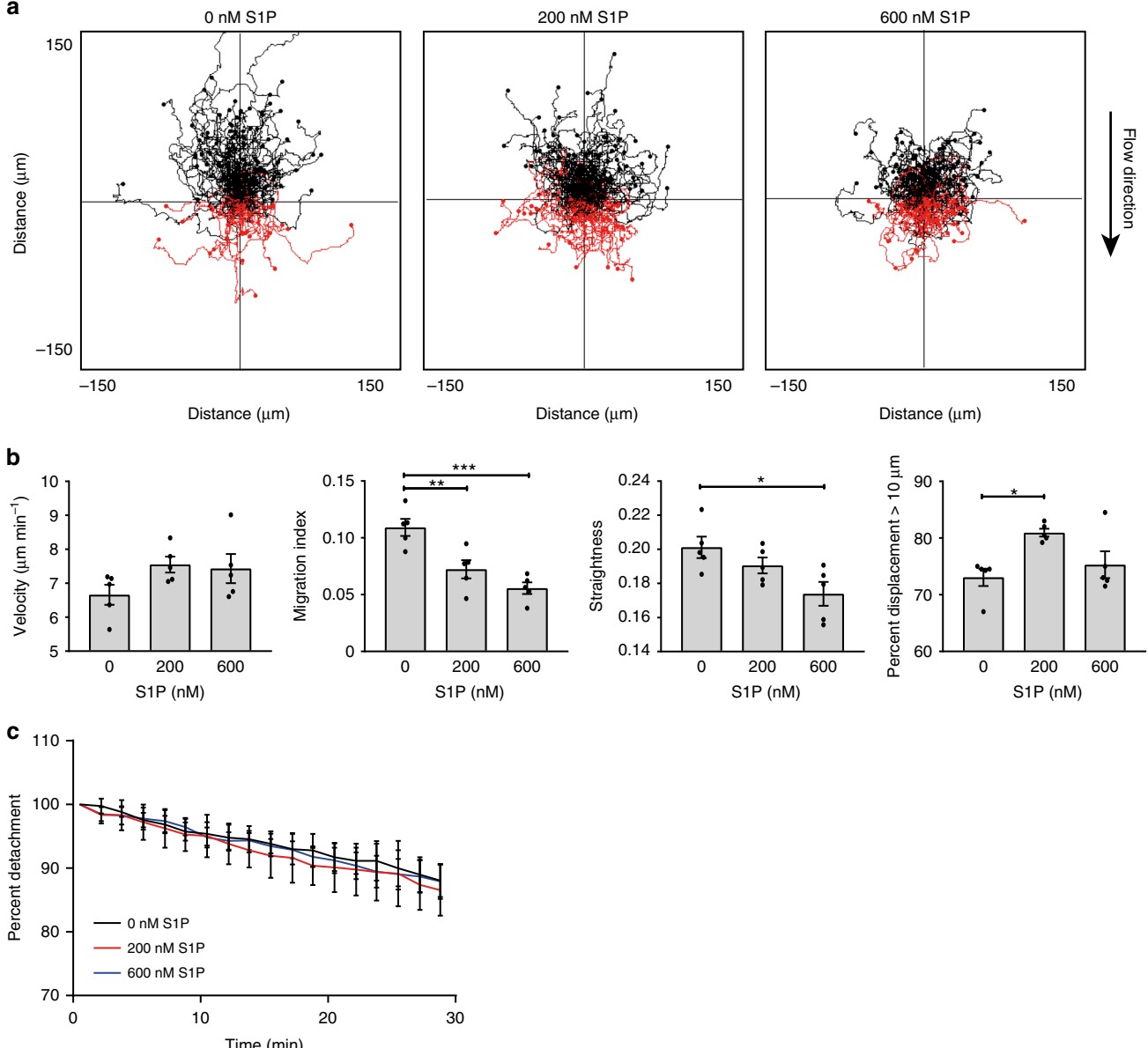

**Fig. 5** S1P inhibits MZB from migrating up shear flow. **a** Representative track plots of MZB migrating under flow (3 dyn cm$^{-2}$) on ICAM-1 (5 µg ml$^{-1}$) treated during migration with 0, 200, or 600 nM of S1P. **b** Quantification of velocity, migration index, straightness, and % of cells that migrated >10 µm. All four graphs show different parameters from the same set of experiments; bars show mean ± SEM. Data are from three experiments: two with two mice each and one with one mouse, total five mice per condition. **c** Detachment of MZB from **a** calculated as a % relative to the number of cells after the start of flow. *$P < 0.05$, **$P < 0.01$, ***$P < 0.001$, one-way ANOVA with Dunnett post hoc tests comparing S1P at 200 or 600–0 nM

and the mutation of *Arhgef6* ($P < 0.0001$), suggesting that both ARHGEF6 and S1P act on integrin adhesion. Although the S1P effect on wild-type cells was relatively modest, an effect that is already detectable in 30 min in an in vitro chamber likely has much stronger consequences in vivo over a time period of days or weeks. Additionally, S1P increased MZB adhesion in a static adhesion assay by ~30–50% (Fig. 6d). Taken together, these findings showed that S1P inhibited the directional migration of MZBs up the flow, presumably by altering their adhesion so that non-motile MZBs became motile but were prevented from migrating up the flow.

**S1PR3 inhibits MZB migration up the flow**. The S1P receptor S1PR3 is critical for MZB migration in a transwell chamber[7,14]. To characterize S1PR3 in shear flow-induced directional

migration, we tested MZBs from wild-type, *Arhgef6*$^{-/-}$, *S1pr3*$^{-/-}$, and *Arhgef6*$^{-/-}$ *S1pr3*$^{-/-}$ double-knockout (dko) mice in the flow chamber. As expected, *Arhgef6*$^{-/-}$ MZBs migrated faster than wild-type cells (Fig. 7a, b). We also observed that S1P modestly inhibited the migration of wild-type MZBs but strongly inhibited the migration of *Arhgef6*$^{-/-}$ MZBs up the flow (Fig. 7a, b). *S1pr3*$^{-/-}$ MZBs were completely resistant to S1P effects although they still express S1PR1, but S1PR1 may be downregulated in the flow chamber due to S1P. Unexpectedly, there was a slight decrease in straightness in the *S1pr3*$^{-/-}$ MZBs that was independent of S1P treatment (Fig. 7a, b). This finding suggested that S1PR3 also has a ligand-independent function in MZBs, potentially in sensing shear stress leading to migration. Strikingly, the strong effect of S1P on *Arhgef6*$^{-/-}$ MZBs was completely abolished in MZBs lacking both *Arhgef6* and *S1pr3* (Fig. 7a, b; Supplementary Movies 1–8). To rule out that any effects we

observed were due to insufficient or excess levels of ICAM-1, we tested the migration of MZBs from all four genotypes on three levels of ICAM-1 (2.5, 5, and 10 µg ml$^{-1}$) (Supplementary Fig. 9a, b). We found no differences among wild-type cells on any of

these levels, but all three genotypes migrated differently on the three ICAM-1 concentrations, therefore it is likely that both mutations affect integrin-mediated adhesion. However, MZBs with a mutation in *Arhgef6* migrate more directionally on

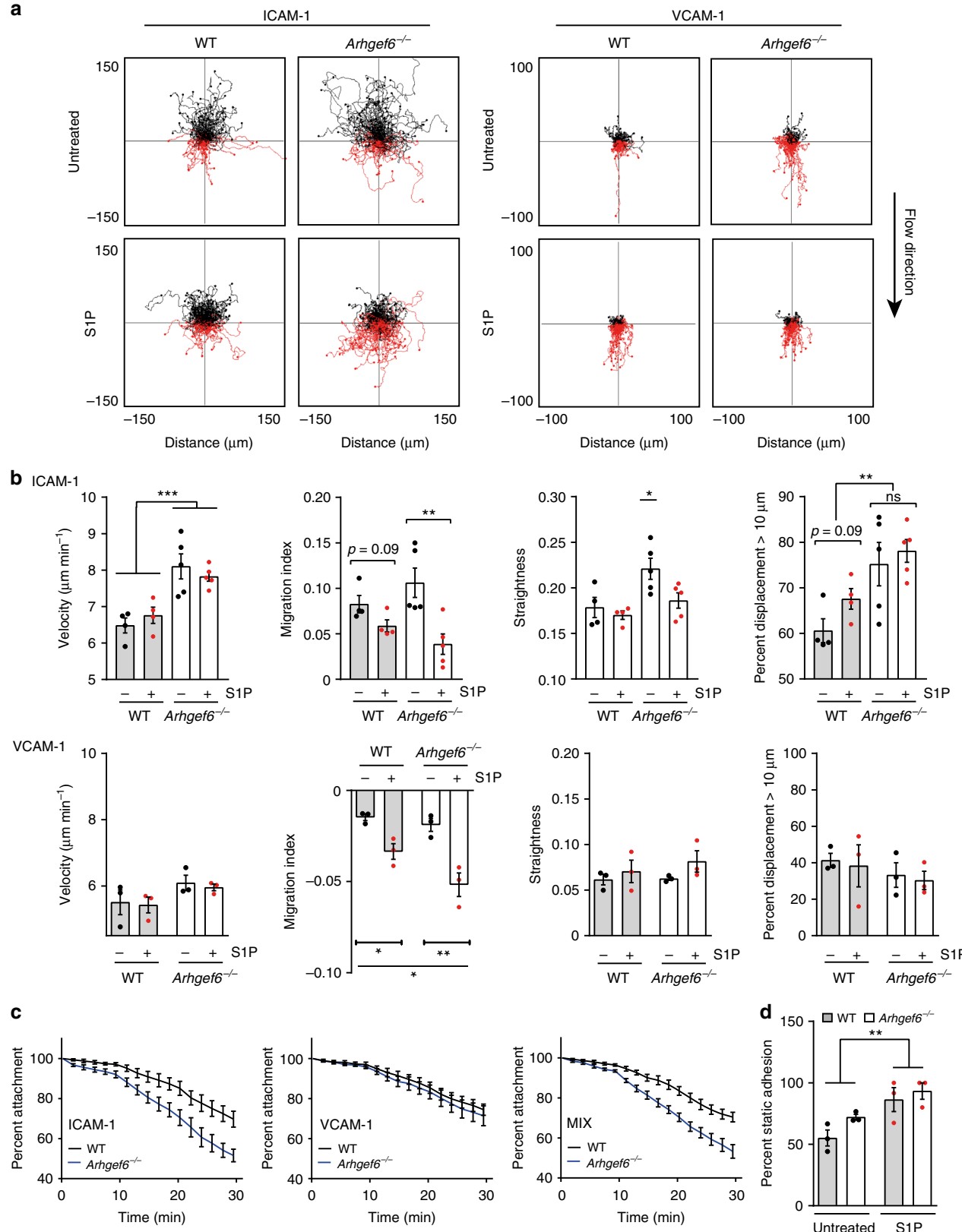

2.5 µg ml$^{-1}$ ICAM-1 than on 5 or 10 µg ml$^{-1}$, while *S1pr3*$^{-/-}$ MZBs migrate less directionally, suggesting that they have opposing effects. Together, these results showed that S1P inhibition of flow-directed migration proceeded via S1PR3.

We next sought to determine how the increased response of *Arhgef6*$^{-/-}$ MZBs to S1P and the deletion of S1PR3 would affect the in vivo localization of these MZBs. To evaluate their positioning, we used the 5-min αCD21 injection method to mark the marginal zone and IgM staining to identify MZBs beyond it. We used bone marrow-reconstituted mice because the *Arhgef6* deletion may affect stroma cell types and *S1pr3*$^{-/-}$ mice have a defective MAdCAM-1$^+$ endothelial layer on the marginal sinus[14], which could potentially affect the shuttling of MZBs into the follicle by making the barrier more porous. MZB numbers from these mice were increased in *Arhgef6*$^{-/-}$ and *Arhgef6*$^{-/-}$ *S1pr3*$^{-/-}$ dko, consistent with the *Arhgef6*$^{-/-}$ phenotype (Supplementary Fig. 6b). In wild-type and *S1pr3*$^{-/-}$ spleens from chimeric mice, no areas of IgM-positive cells were located just outside the border of the marginal zone (patches of strong green staining in the middle of the red pulp likely represent plasma B cells) (Fig. 8a, b). In *Arhgef6*$^{-/-}$ spleens, large areas of IgM-positive MZBs were found outside the CD21-marked zone (Fig. 8a, b), reflecting the defect in adhesion of *Arhgef6*$^{-/-}$ MZBs observed in the detachment under flow assay (Fig. 6c). A closeup of spleen sections from *Arhgef6*$^{-/-}$ mice revealed that the *Arhgef6*$^{-/-}$ MZBs were interspersed with F4/80$^+$ red pulp macrophages (Supplementary Fig. 4b). Consistent with the idea that some *Arhgef6*$^{-/-}$ MZBs are washed out of the marginal zone by the force of blood flow, there was a slight increase in *Arhgef6*$^{-/-}$ MZBs found in the peripheral blood (Supplementary Fig. 6c). As predicted by the finding that the deletion of *S1pr3* reversed the effects of S1P on *Arhgef6*$^{-/-}$ MZB flow migration, spleen sections of *Arhgef6*$^{-/-}$ *S1pr3*$^{-/-}$ dko mice showed that the mislocalization of *Arhgef6*$^{-/-}$ MZBs was reversed by the additional deletion of *S1pr3*. These results reveal that S1PR3 exerts a "downward" directional force on MZBs toward the red pulp and that deleting *S1pr3* counters the S1P effect and allows MZBs to respond to flow by migrating toward the follicle (see Fig. 9 for model).

## Discussion
Here, we demonstrated that shear flow induces marginal zone B cells (MZB) adhered to intercellular adhesion molecule 1 (ICAM-1) to migrate toward the source of flow. This finding was surprising because it might be predicted that the haemodynamics of blood in the spleen would induce adhesion by integrins on MZBs to keep MZBs in the marginal zone. In fact, the opposite is true: shear flow induces migration from the marginal zone toward the follicle. We also showed that sphingosine-1-phosphate (S1P) signaling provides a counterforce to inhibit this migration (see model, Fig. 9). It is known that MZB shuttling between the marginal zone and the follicle requires S1PR1 and CXCL13, which exert opposing effects; S1PR1 is required for MZBs to exit the follicle, while the CXCR5-CXCL13 pathway is required for

MZBs to enter the follicle[13,28]. Our results add to this model by introducing shear flow as an inducer of directional migration in the marginal zone toward the follicle. A gradient of CXCL13, either soluble or surface bound, would be difficult to establish in the blood flow through the marginal zone. It is more likely that CXCL13 fixed on the mucosal vascular addressin cell adhesion molecule 1 (MAdCAM-1)$^+$ endothelial layer of the sinus enhances haptotactic migration through this cell layer into the follicle[20,29]. We found that CXCL13 and MAdCAM-1 inhibited MZB migration up the flow, supporting the idea that they provide assistance to MZB shuttling into the follicle at the point where flow is strongest, the sinus, by inducing strong adhesion. The marginal sinus is no more than several cell widths wide, ensuring that MZBs contact the MAdCAM-1$^+$ CXCL13$^+$ endothelial surfaces before migrating too far into the capillaries that feed the sinus. Thus, for MZBs, shear flow induces directed migration in the same manner as chemokines, while the actual chemokine CXCL13 may serve as a turnstile at the endothelial barrier to the follicle.

To study the movement of MZBs in vivo, we used *Arhgef6*$^{-/-}$ MZBs. One defining feature of *Arhgef6*$^{-/-}$ lymphocytes is increased speed[24]. Interestingly, this phenotype may also explain the elevated MZB numbers in *Arhgef6*$^{-/-}$ mice: increased *Arhgef6*$^{-/-}$ T-cell speed is linked to defective cell–cell contacts[24], and MZB cell–cell contacts with invariant natural killer T (iNKT) cells are required to inhibit MZB numbers[30]. Thus, increased *Arhgef6*$^{-/-}$ MZB speed may reduce contacts with iNKT, leading to increased numbers. Increased cell speed is also inversely correlated with adhesion[25–27], and both ARHGEF6 and its homolog ARHGEF7 (aka βPIX) have been implicated in integrin functions[31]. Integrins mediate adhesion, and *Arhgef6*$^{-/-}$ MZBs have normal static adhesion but defective adhesion under flow. Additionally, *Arhgef6*$^{-/-}$ MZBs were strongly inhibited from migrating up the flow by S1P, to an even greater extent than wild-type MZBs were. This S1P-induced down-flow movement of *Arhgef6*$^{-/-}$ MZBs, coupled with increased detachment under flow and increased downward migration on vascular cell adhesion protein 1 (VCAM-1), acted to lodge *Arhgef6*$^{-/-}$ MZBs in the red pulp. In low-flow areas of the red pulp, where the reticular meshwork is dense, the *Arhgef6*$^{-/-}$ MZBs would adhere normally and would therefore remain mislocalized in the red pulp. These effects were reversed in *Arhgef6*$^{-/-}$ *S1pr3*$^{-/-}$ dko MZBs, showing that S1PR3 is the principal S1P receptor for counteracting migration up the shear flow.

The migration of MZBs up the flow was only possible when ICAM-1 was present and the proportion of VCAM-1 did not exceed 50%, showing that the corresponding integrins, lymphocyte function-associated antigen 1 (LFA-1), and very late antigen 4 (VLA-4), have qualitatively different functions. These findings reveal that LFA-1 is required for maintaining MZBs in the spleen and for migration up the flow. The results also suggest that VLA-4 may act as a backup adhesion system for MZBs under flow. Low VCAM-1 resulted in minimal, non-directional migration, while high VCAM-1 or high flow resulted in the MZBs moving down

**Fig. 6** *Arhgef6*$^{-/-}$ MZB migrate faster, detach more under flow, and respond strongly to S1P. **a** Representative track plots of wild-type (WT) and *Arhgef6*$^{-/-}$ MZB bells, untreated or with S1P (200 nM), migrating under flow (3 dyn cm$^{-2}$) on ICAM-1 (2.5 µg ml$^{-1}$) or VCAM-1 (2.5 µg ml$^{-1}$). **b** Quantification of migration index, velocity, straightness, and percentage of cells that migrated >10 µm for WT (gray bars) and *Arhgef6*$^{-/-}$ (white bars) MZB on ICAM-1 or VCAM-1 (WT: n = 4 mice, *Arhgef6*$^{-/-}$: n = 5 mice; 5 experiments total). Data are expressed as mean ± SEM. *P < 0.05, **P < 0.01, ***P < 0.001, by t test (for migration index and % displacement on ICAM-1), one-way ANOVA (for straightness and % displacement, comparison of values to WT), and two-way ANOVA (velocity on ICAM-1, migration index on VCAM-1). **c** Detachment of WT and *Arhgef6*$^{-/-}$ MZB. Fraction of remaining cells calculated as a % relative to number of cells after start of flow on ICAM-1 (1.5 µg ml$^{-1}$), VCAM-1 (1.5 µg ml$^{-1}$), or ICAM-1 and VCAM-1 together (0.75 µg ml$^{-1}$ each) under flows of 3 dyn cm$^{-2}$ (0–10 min), 9 dyn cm$^{-2}$ (10–20 min), and 15 dyn cm$^{-2}$ (20–30 min) (n = 5 mice per genotype, 5 experiments total). **d** Static adhesion of WT and *Arhgef6*$^{-/-}$ MZB treated with S1P (200 nM) on ICAM-1 (2.5 µg ml$^{-1}$), relative to total adhesion on poly-lysine. Bars are mean plus SEM. **P < 0.01, two-way ANOVA for S1P treatment difference. Data are from three experiments with two mice (one per genotype) each. ns not significant

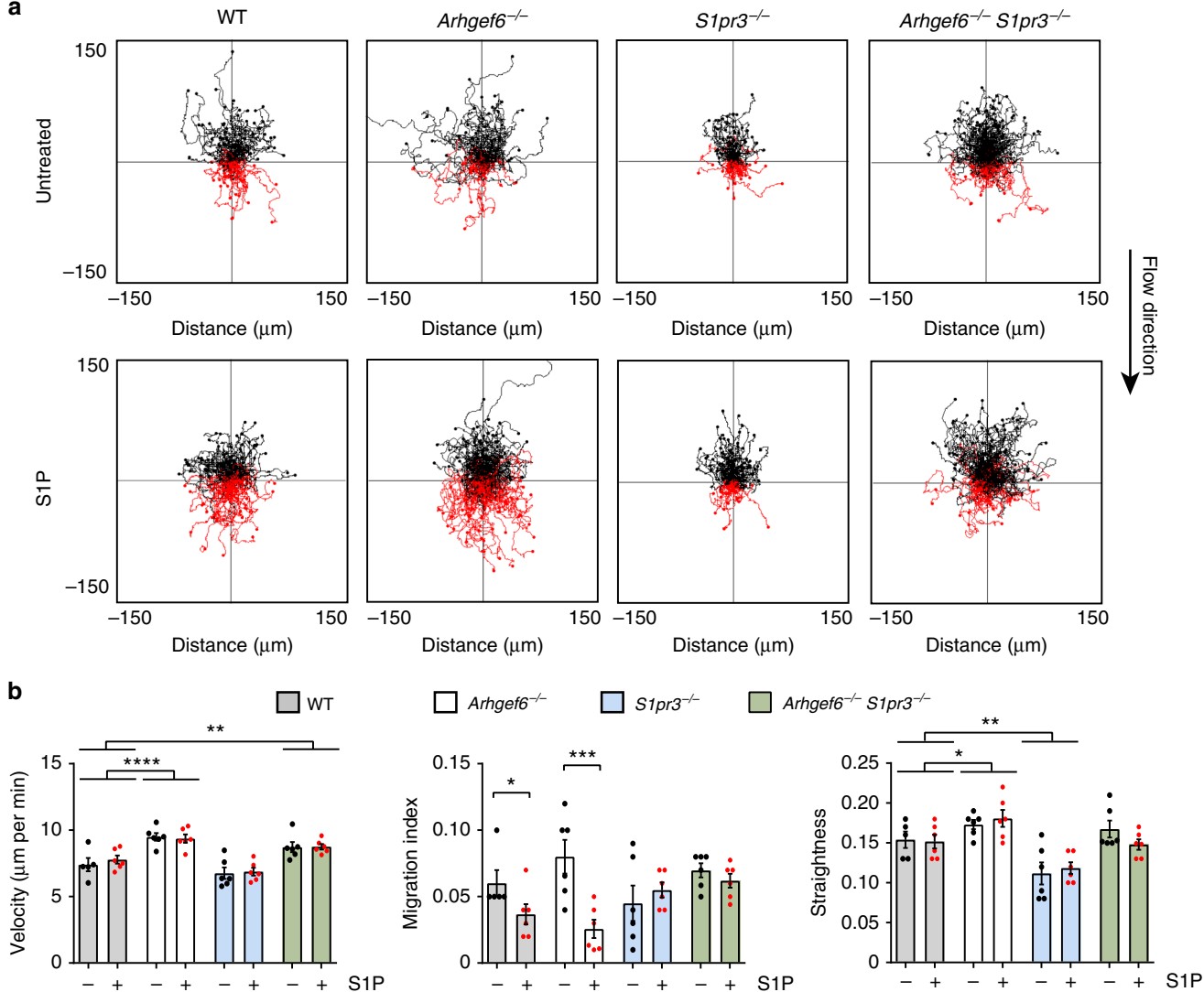

**Fig. 7** S1P inhibition of MZB migration up the flow requires S1PR3. **a** Representative track plots of wild-type (WT), *Arhgef6*$^{-/-}$, *S1pr3*$^{-/-}$, and *Arhgef6*$^{-/-}$*S1pr3*$^{-/-}$ MZB, untreated or treated with S1P (200 nM), migrating under flow (3 dyn cm$^{-2}$) on ICAM-1 (2.5 μg ml$^{-1}$). **b** Quantification of velocity, migration index, and straightness. Two-way ANOVA significance indicated with horizontal bars for genotype significance, brackets for Mann–Whitney test (migration index). Bars show mean ± SEM. *$P < 0.05$, **$P < 0.01$, ***$P < 0.001$, ****$P < 0.0001$. Data represent 12 experiments with two mice each, six mice per genotype total. Symbols in one condition group denote the replicates and represent the average of 100–200 cells from each individual mouse

the flow, as if being passively propelled by shear force. The marginal zone consists mainly of ICAM-1, with a smaller proportion of VCAM-1. When shear flow and ICAM-1 are present in the marginal zone, MZBs migrate well up the flow, suggesting that LFA-1 is distributed to position the MZB along the axis of flow. However, if the flow direction were to change, the MZBs would need to re-position LFA-1. At this moment, MZBs without active VLA-4 would be vulnerable to being washed out by the flow and ending up in the VCAM-1-rich red pulp. VCAM-1 is expressed at high levels on red pulp macrophages, which are often clustered around venous sinuses in the red pulp; thus, it is possible that an additional VLA-4 function may be to ensure the disposal of non-functional MZBs through phagocytosis.

S1P receptors are an important target for immunosuppressive drugs because they promote lymphocyte exit from lymph nodes by counteracting chemokine signaling[10,32–34]. This role of S1P in mobilizing lymph node T cells into circulation is an interesting contrast to the role we have identified for S1P in maintaining MZB positioning in the marginal zone. For example, S1P signaling through S1PR1 opposes the actions of CCR7 for T cells[35],

and S1P signaling through S1PR2 opposes the actions of CXCL13 for germinal center (GC) B cells[36]. However, if we consider shear force-induced directional migration as analogous to chemotaxis, then it becomes clear that S1P acts similarly in MZB positioning by opposing flow-induced migration. Why would S1P impede the migration of MZBs toward the follicle, when shuttling antigens picked up from blood into the follicle is their job? One possible reason is that it increases transit time in the marginal zone so that MZBs have a better chance of being exposed to potential antigens in the blood. Similarly, S1P may act to increase productive contacts between MZBs and other immune cells in the marginal zone, such as macrophages, NKT cells, and neutrophils[37–39]. In the present findings, S1P-treated MZBs are less capable of migrating up the flow than untreated cells but they are just as fast. Therefore, the net result would be lateral migration within the confines of the marginal zone, allowing the MZBs to interact with other immune cells until the S1P receptors become internalized, at which point the MZBs would be free to migrate up the flow toward the follicle. Although S1PR1 is rapidly downregulated on MZBs exposed to S1P, S1PR3 may also be downregulated, albeit

## a

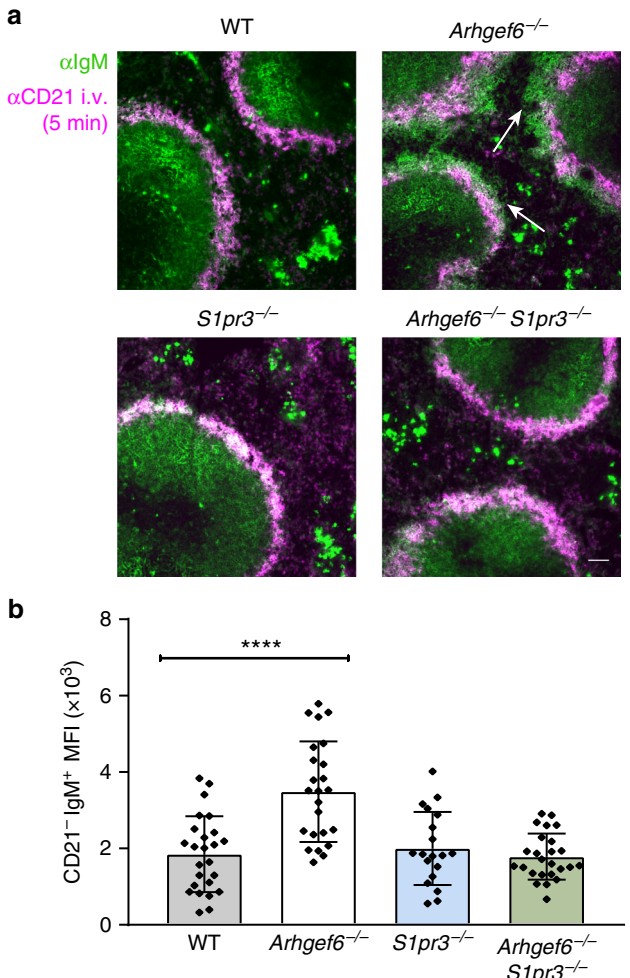

**Fig. 8** Translocation of αPIX knockout MZB toward the red pulp is rescued by co-deletion of S1PR3. **a** Immunofluorescence microscopy of IgM⁺ cells located beyond the CD21-positive boundary of the marginal zone (white arrows) in wild-type (WT), Arhgef6⁻/⁻, S1pr3⁻/⁻, and Arhgef6⁻/⁻S1pr3⁻/⁻ chimeric spleens. Scale bar, 50 μm. **b** Quantification of mean fluorescence intensity (MFI) of CD21⁻ IgM⁺ cells inside a band of 60 μm beyond the marginal zone, itself calculated as a band of 60 μm outside of MAdCAM-1 staining. Bars show mean ± SEM. Data represent two independent experiments with three bone marrow chimeric mice per genotype. Symbols in one condition group denote individual follicles (WT, Arhgef6⁻/⁻ and Arhgef6⁻/⁻S1pr3⁻/⁻: n = 5 mice; S1pr3⁻/⁻: n = 4 mice). ****P < 0.0001, one-way ANOVA with Dunnett post hoc test relative to wild type

more slowly, since the MZBs that need to migrate up the flow toward the follicle might not otherwise be able to overcome the drag force of S1PR3 signaling. In support of this explanation, there is evidence for ligand-induced internalization of S1PR3 in other cell types[40,41]. We also observed an unexpected ligand-independent function for S1PR3 on MZBs. It is possible that shear flow activates S1PR3, as it activates S1PR1 on endothelial cells, even when S1P binding is not possible[42]. In summary, these findings establish haemodynamics and ICAM-1 as a new force that guides MZBs to the follicle and that is countered by VCAM-1 and S1P signaling through S1PR3.

## Methods

**Mice**. Arhgef6⁻/⁻ mice were previously described[23]. Mice were housed in specific-pathogen-free conditions according to institutional guidelines. All procedures were performed in accordance with the institutional guidelines for health and care of experimental animals and were approved by the Landesverwaltungsamt Halle

(representing the state of Saxony-Anhalt), Germany and Landesamt für Gesundheit und Soziales Berlin. S1pr3⁻/⁻ mice were a gift from J. Chun[43]. Both strains were backcrossed to C57BL/6 for over 20 generations. Mice were killed by CO₂ inhalation for 10 min or by cervical dislocation. For bone marrow chimeric reconstituted mice, C57Bl/6 Ly5.1 8–12-week-old male and female host mice were lethally irradiated at 10 Gy (BioBeam 8000 (Gamma Service Medical GmbH, Germany) providing gamma irradiation (¹³⁷Cs)), then injected intravenously with 2 × 10⁶ bone marrow cells derived from sex- and age-matched Ly5.2 donor mice. Recipient mice were immobilized by anesthesia with intraperitoneal injection of 100 μl ketamine (10 mg ml⁻¹) and xylazine (2 mg ml⁻¹) solution. Ly5.1 recipient mice (six per genotype) were reconstituted with bone marrow derived from two independent donors and analyzed 8–12 weeks later.

**Flow cytometry and MZB purification**. MZBs were stained using antibodies from BD Bioscience, Germany (B220-V450, CD23-BV510, CD43-PE, CD1d-FITC, or −BV421), Biolegend (CD21-APC and CD21-FITC, IgM-Pe-Cy7, Ly5.2-APC, CD11b-APC-Cy7), or Miltenyi (IgM-APC). For testing integrin expression on MZB, purified cells were analyzed by flow cytometry using BD Bioscience antibodies anti-α4-FITC (R1-2), anti-αL-PE (2D7), anti-β2-FITC (C7116), and anti-β1-FITC (HMB1-1, Santa Cruz). To determine MZB numbers in peripheral blood, heparinized samples were stained for surface expression of B220, CD1, CD21, TCRβ-APC-Cy7 (Biolegend), and Ter119-APC-Cy7 (BD Bioscience). FACS data were acquired and analyzed using a BD FACSCanto II flow cytometer, BD FACSDiva software, and FlowJo (Tree Star).

For MZB purification by flow cytometry, splenocytes from male and female age-matched wild-type and knockout mice were purified by FACS sorting (FACS Aria III (BD Bioscience) gating on CD43⁻ CD21⁺ B cells and on CD21ʰⁱ CD23ˡᵒʷ MZB. For MZB purification by MACS columns, MZB were purified using a negative B-cell isolation kit for MZB (Miltenyi) that keeps MZB unstained. In both cases, erythrocytes were lysed in a standard NH₄Cl/KHCO₃/EDTA lysis buffer at room temperature for 2 min after spleen dissociation. Cells were subsequently kept cold for all remaining steps of the purification, and their migration potential was analysed immediately in a flow chamber. For MZB preparation using the kit, cells were purified in two negative isolation steps over columns in an AutoMacs (Miltenyi). The initial step of removing CD43⁺ non-B cells from the cell suspension included an antibody against Ter119 to remove any remaining erythrocytes from the suspension. During the isolation, cells were maintained and washed in PBS containing 0.4% fatty acid-free BSA and were never exposed to serum or any other BSA. The final purity of the B220⁺ CD21ʰⁱ CD23ˡᵒ MZB was over 90%. For testing viability of MZB and FOB cells, purified cells were incubated in PBS/2% BSA followed by staining with 7-aminoactinomycin D (7-AAD) and Annexin V FITC (BD Biosciences) and analyzed by flow cytometry.

**Flow chamber migration assays**. Ibidi flow chamber slide (μIV-0.4) (Ibidi, Martinsried, Germany) channels were coated overnight at 4 °C with indicated amounts of ICAM-1 or VCAM-1 (R&D Systems), blocked for 1 h at room temperature with 2% fatty acid-free BSA (Roth, Germany) in PBS, then washed 2× in PBS and filled with migration medium (HBSS containing 0.8 mM Mg²⁺, 1.26 mM Ca²⁺, 0.4% BSA, 10 mM HEPES). In some experiments, slides were coated with MAdCAM-1, ICAM-2, CXCL13, or CXCL12 (R&D). For S1P treatment, S1P (Caymen Chemical) was resuspended in methanol, sonicated, then dried under a stream of nitrogen gas. Aliquots were resuspended in PBS + 0.4% BSA, then added to migration buffer in the fluidics unit of the Ibidi pump system at a concentration of 200 nM unless otherwise indicated. Purified MZB (1.5 × 10⁶ per ml) were added to channels and incubated at 37 °C for 30 min to adhere and to induce re-sensitization of S1P receptors before tubing assembly was attached to the slides and to an Ibidi pump system filled with pre-warmed migration medium. Images were acquired at ×10 on a Leica DMI 6000 inverted wide-field microscope. Cells were tracked using the ImageJ plugin Mtrack2. Migration parameters were calculated using the Ibidi Chemotaxis tool. The migration index was calculated as the average of the ratios of the projected y-coordinates of a cell's end point to its track length. Straightness was calculated as the ratio of the cell's path length to its track length. Velocity was calculated as the ratio of a cell's track length to elapsed time.

**MZB static adhesion and transwell migration**. Purified MZ B cells were incubated at 37 °C for 1 h in 96-well plates coated with ICAM-1 (10 μg ml⁻¹, R&D Systems). Poly-L-lysine (0.01%, Sigma)-coated wells were used as a measure of input. Non-adherent cells were dislodged by gently inverting the assay plates for an additional 30 min. Attached cells were fixed in the wells with the addition of 4% (w/v) formaldehyde for 15 min at room temperature. Adherent cells were quantified with crystal violet staining and absorbance reading in a Tecan Infinite M200 plate reader. Background crystal violet staining was subtracted and the percentage of attached cells was determined by comparison with the number of adherent cells found in poly-L-lysine-coated wells (set to 100%). For transwell migration assays, 5 μm pore size membranes (Corning Costar) were coated with 10 μg ml⁻¹ recombinant mouse ICAM-1/CD54 Fc (R&D Systems) and blocked with 2% BSA. Migration of cells to S1P and CXCL13 (R&D) was analyzed as described (Korthals et al.[24]).

**Mouse treatments**. For treatment with anti-LFA-1, mice were injected with either control IgG or with 100 μl (100 μg) anti-LFA-1 (M17/4) (Affymetrix, eBioscience; 16–0111) 30 min before double in vivo labeling. For double in vivo labeling, mice were intravenously injected with 1 μg of αCD35-bio, followed 15 min later (unless stated otherwise) with 1 μg αCD21-PE. Five minutes later, mice were sacrificed, spleens were rapidly removed, and dissected for histology. Microscopy of spleen sections showed MZB in follicles that were singly labeled with the first antibody (αCD35) but not the second (αCD21), called "shuttlers," indicating that during the time interval between injections, the cells left the marginal zone, which is accessible to the blood-borne labeled-antibody, to the follicle, which is inaccessible to the

antibody[8]. For spleen excision surgery, mice were anaesthetized for terminal procedures by intraperitoneal injection with 100 μl PBS containing ketamine (10 mg ml$^{-1}$) and xylazine (2 mg ml$^{-1}$) and placed on a warming pad, followed by an injection of 800 U kg$^{-1}$ heparin 15 min before start of surgery. The procedure was initiated with an i.v. injection of αCD21-PE, followed by an incision made below the costal margin in the left flank, the abdominal cavity was entered, and the spleen examined. At 5 min after injection, simultaneous surgical ligation of all splenic vessels was performed at the hilum and the spleen was excised. For control animals, the spleen remained in situ in the peritoneal cave. The excised spleens were incubated in a dish containing 37 °C oxygen-perfused medium (RPMI; Gibco), as

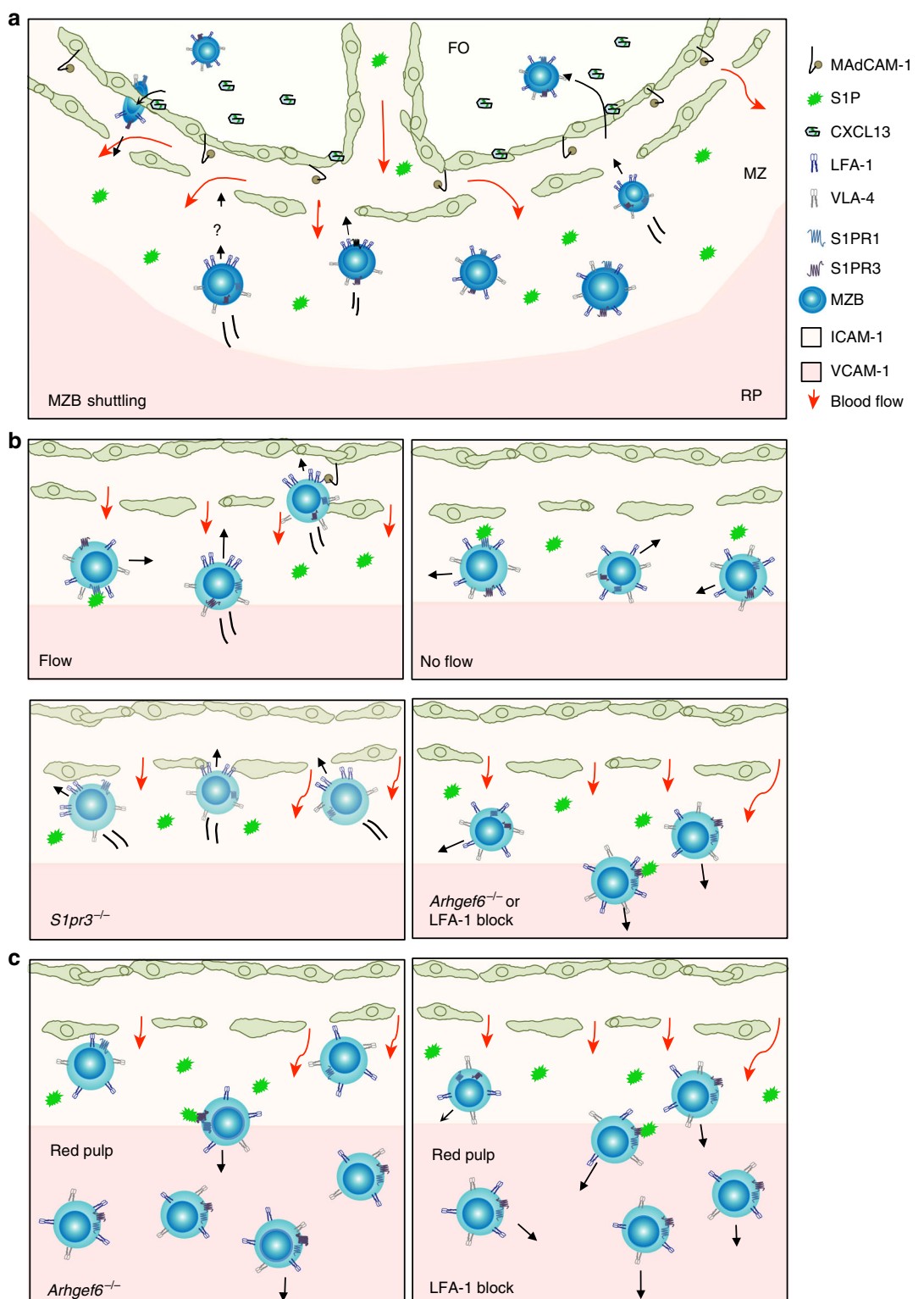

**Fig. 9** Model of MZB migration up flow with S1P counterforce. **a** MZBs in spleen shuttle from the marginal zone (MZ) surrounding the follicle (FO) to the inside of the follicle, to carry antigens from blood to the follicular dendritic cells and back again. CXCL13 is required for MZB entry into the follicle, while S1PR1 is required for MZB exit from the follicle. The marginal zone is rich in ICAM-1 while the red pulp contains both ICAM-1 and high levels of VCAM-1. An unknown force drives MZBs toward the follicle. **b** Flow: Shear flow induces MZB migration up the flow if ICAM-1 is present. S1P signaling through S1P receptors inhibits MZB migration up flow and may trigger LFA-1 integrin re-distribution. S1PR1 is rapidly internalized, thus S1PR3 may be the main counterforce. Upon eventual internalization of S1PR3, MZBs would migrate up the flow. MZBs that reach the sinus (elongated green cells) would bind strongly to MAdCAM-1 and CXCL13, preventing their migration against flow but protecting them from washing out. MZBs would then transmigrate through the sinus into the follicle. No flow: without flow, MZBs migrate for shorter distances in random directions and MZB follicular shuttling is impaired. $S1pr3^{-/-}$: without S1PR3, MZBs do not respond to S1P and are not inhibited from migrating up the flow toward the follicle. $Arhgef6^{-/-}$ or LFA-1 block: MZBs that undergo a loss of adhesion, through $Arhgef6$ mutation or LFA-1 blockade, are susceptible to washing out of the marginal zone toward the red pulp. S1P binding to receptors would accelerate this process. Impaired migration up flow causes reduced shuttling into the follicle. **c** VCAM-1 in the red pulp. MZBs can adhere to VCAM-1 under flow but do not migrate on it. MZBs with reduced adhesion via LFA-1 are conveyed by flow to the red pulp. Left panel: $Arhgef6^{-/-}$ MZBs that reach the red pulp can adhere to it and resist being washed out by flow. Right panel: MZBs with blocked αLFA-1 are only able to partially resist the force of the flow by adhering to VCAM-1

previously reported[44]. To perfuse, a tube from a peristaltic pump (Gilson Minipuls 3) fed by a canister of 95% $O_2$/5% $CO_2$ was placed in the dish in such a manner as to avoid creating a stream or flow. After 25 min, spleens were frozen for histology.

**Histology**. For histology of spleen sections, spleens were rapidly embedded in Tissue Freezing Medium (Leica) and shock-frozen in isopenthane (−40 to −50 °C). Sections (12 μm) were cut with a cryostat, air-dried, and fixed with 2% PFA for 10 min. Sections were pre-incubated for 10 minute in 10% normal donkey serum/0.1% NaN3 in PBS, then incubated overnight at 4 °C with rabbit-anti-PE (1:500) (Rockland 200–4199) and streptavidin-Alexa Fluor 488 (1:200) (Molecular Probes S11223) in 10% normal donkey serum/0.1% NaN3 in PBS. Sections were then washed 3× for 10 min each in PBS, pre-incubated in 0.1% BSA in PBS for 10 min, incubated for 1 h with Cy3-conjugated donkey-anti-rabbit (1:500) (Dianova 711–165–152) in 0.1% BSA in PBS, washed in PBS for 3× for 10 min each, and mounted in Immu-Mount (ThermoFisher). For VCAM-1/ICAM-1 staining, sections were stained as above with rabbit-anti-VCAM-1 (1:50) (Santa Cruz, H-276) or rat-anti-mouse CD106 (1:50; Biolegend 105702) and goat-anti-mouse ICAM-1 (1:200; R&D AF796) followed by Cy3-conjugated donkey-anti- rabbit (1:500) and Alexa 488-conjugated donkey-anti-goat. For IgM staining, sections from mice injected with CD21-PE (PE Rat anti-mouse CD21/CD35 clone 7G6, BD) alone or co-injected with F4/80 Alexa Fluor 488 (Biolegend) for 5 min were pre-incubated for 10 min in 10% normal donkey serum/0.1% NaN3 in PBS, then incubated overnight at 4 °C with rabbit-anti-PE as above and Alexa Fluor 647 anti-mouse IgM (1:500; Biozol) in PBS, washed 3× for 10 min each in PBS, then mounted in Immu-Mount. Slides were visualized using a Leica DMI 6000 inverted wide-field microscope with a ×10 objective and processed with Leica Application Suite software.

**Quantification of MZB in spleen histology**. Spleen sections with injected αCD21-PE were co-stained as described in figure legends for IgM and MAdCAM-1 (mouse MAdCAM-1 antibody goat IgG, 1:25, R&D and Alexa Fluor 647 donkey anti-goat, ThermoFisher), and in some cases, CD169 (1:50; Alexa Fluor 647 anti-mouse CD169 Clone 3D6.112), and relevant cells were quantified by ImageJ analysis of mean gray intensity in selected areas. To quantify MZB in the red pulp in the LFA-1-blocking experiment, a line was drawn on the outside border of CD21 staining, and a band of 35 μm was made from the line in ImageJ. To quantify MZB in the red pulp in the $Arhgef6^{-/-}$ and $S1PR3^{-/-}$ experiment, the marginal zone was delineated using a 70 μm band with an inner border line drawn on the marginal sinus using MAdCAM-1 and CD169 staining. An additional 70 μm band was drawn outside this marginal zone band to capture IgM signals in the red pulp adjacent to the marginal zone. To quantify shuttlers inside the follicles, a line was drawn over the sinus using MAdCAM-1 and CD169 staining to delineate the outer border of a smaller, 70 μm band that was drawn inside the follicle. In all cases, if non-specific fluorescence from adjacent follicles or from IgM+ plasma cell staining in the red pulp appeared in the bands, these were removed before calculating mean gray intensity minus background values.

**Statistical analysis**. All statistical tests were performed with Graphpad Prism 6. For wild type alone, statistical analyses were made with one-way ANOVA with Dunnett post hoc test performed relative to initial value. For more than one genotype, two-way ANOVA was used with Sidak post hoc tests. $t$ tests were used where indicated in figure legends. To avoid pseudoreplication, the values of 100–200 cell tracks from one mouse were averaged per experiment. Statistically significant differences are indicated on the figures according to this legend: *$P <$ 0.05, **$P <$ 0.01, ***$P <$ 0.001, ****$P <$ 0.0001.

**Data availability**. Data supporting the findings of this study are available in the article and its Supplementary Information files and are available from the corresponding authors on request.

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

## Acknowledgements

We thank H. Baumann, A. Mohrmann, and C. Schwarzer for technical assistance, and members of the Fischer laboratory for discussions. We also thank R. Hartig for FACS cell sorting and J. Chun for providing S1PR3 knockout mice. This work was supported by grants of the "Deutsche Forschungsgemeinschaft" SFB 854/TP11 to K.-D.F., and TRR130, TP17, and C01 to A.E.H.

## Author contributions

Conceptualization: K.T. and K.-D.F.; methodology: K.T., M.S., S.K., M.K., A.E.H., and K.-D.F.; investigation: K.T., M.S., S.K., K.R., L.T., M.K., Y.E, I.J., and K.S.; formal analysis: K.T. and M.K.; writing manuscript: K.T.; writing review and editing: K.T and K.-D.F.; funding acquisition: K.-D.F. and A.E.H.; supervision: K.T., M.K., A.E.H., and K.-D.F.

## Additional information

**Competing interests:** The authors declare no competing financial interests.

