## [Peer Review File · Nature Communications]

Reviewers' comments:

Reviewer #1 (Remarks to the Author):

The manuscript by Tedford et al adds to the mechanism on how Marginal Zone B cells shuttle in the spleen to carry antigen to the follicle to support an adaptive immune response. This movement has been studied previously by The Cyster lab that defined that the migratory pattern relied on GRK2 mediated S1PR1 desensitization and resensitization. Here this is investigated further to include migration against flow. I think the study is interesting but have a couple of concerns that I think should be addressed with experiments or in writing.

Major concerns:

- In the studies on VCAM and ICAM am lacking the control of not having any adhesion molecule there or an irrelevant protein.
- Which is the dominating force when comparing S1P1 and S1PR3 and how big is the expression difference? Also, the mechanism which by SiPR3 responds to flow should be further substantiated.
- Does the knocking out of aPIX alter the expression of receptors needed for transport of TNP-Ficoll by MZBs?

Minor concerns:

- The flow of the spleen is often described as "turbulent current". However, the blood flow slows down significantly in the spleen due to the much larger volume when entering. Thus the overall description of the flow in the spleen could be discussed better.
- This study would greatly be helped by adding a more complete mechanism picture to better explain the different forces affecting MZB migration.
- What was the purity and viability of the sorted MZBs and FOBs?

Reviewer #2 (Remarks to the Author):

In the manuscript "The opposing forces of shear flow and sphingosine 1-phosphate control marginal zone B cell shuttling", the authors use an elegant in vitro system to characterize the movement of MZB cells under shear forces. They show that ICAM-1 promoted migration of MZB cells towards the flow, while VCAM-1 enhanced adhesion and promoted movement in the

opposite direction. Histological analysis showed that ICAM-1 was highly expressed in the marginal zone, while VCAM-1 was abundant in the red pulp. S1P signaling through S1P3 was suggested to counteract migration against flow, potentially leading to movement towards the red pulp. In addition, MZB cells derived from α PIX Rho-GEF KO mice showed increased motility under shear force, a defect that could be partially corrected when crossed with S1P3 KO mice. The authors also attempted to assess the distribution of MZB cells in mice in which α PIX Rho-GEF, S1P3 or LFA-1 function is impaired. They propose that MZB cell distribution is shifted towards the red pulp when α PIX Rho-GEF or LFA-1 are dysfunctional and that this phenotype depends largely on signaling via S1P3.

The model suggested by the in vitro data in this manual script is compelling, complements previous studies and generates new interesting hypotheses. Overall, the observations described in this work are novel and have the potential to stimulate new thoughts and exciting scientific discussion in the immunology and cell migration fields. However, we feel that in its current form, the manual script suffers from insufficient strength to support some of its main conclusions and can greatly benefit from an extensive revision. In addition, the in vivo data are not convincing and the authors should consider removing them altogether.

Major concerns:

1. The main claim of the paper is that MZB cells migrate in opposite directions in response to S1P and shear flow. This statement is largely based on quantification of cell velocity, track straightness and migration index under various conditions. However, direct quantification of directionality (specifically, movement up or down the flow) is missing. Such analysis will greatly improve the strength of the arguments.
2. Figure 2, the authors claim that VCAM-1 expression is restricted to the red pulp and that in the absence of ICAM-1 it does not support migration against flow. These observations seem at odds with published works showing that expression of VCAM-1 alone is sufficient to allow retention in the marginal zone. Is it possible that VCAM-1 is expressed in lower levels in the marginal zone region and that its ability to enhance adhesion (as suggested by the authors) under flow is sufficient to promote retention in the absence of ICAM-1? Better histology may help to clarify this point. In addition, to further clarify the relationship between ICAM-1 and VCMA-1 in promoting directional migration of MZB cells, it will be useful to repeat the experiment described in figure 2a using a constant concentration of ICAM-1 mixed with increasing concentrations of VCAM-1 (an irrelevant protein can be added to maintain constant total protein density across the chamber's surface).
3. Figure 3- the authors show that detachment of α PIX MZB cells from VCAM-1 under flow is similar to WT cells. Surprisingly, addition of ICAM-1 led to significantly increased detachment

from VCAM-1. The physiological significance of these observations is not clear and it is somewhat surprising that reduced adhesion in vivo does not lead to gradual loss of the marginal zone compartment in α PIX KO. Comparing detachment of α PIX KO MZB and FOB cells may help to clarify this point.

4. In figure 4 the authors claim that S1P promotes adhesion and motility but inhibits directional migration of MZB against the flow. However, the effect induced by S1P treatment on WT is not convincing and the absence of quantitative analysis of directionality (e.g. how many cells go up or down the flow?) makes it difficult to interpret the results.

It is possible that the relatively modest effect of S1P in these assays can be improved if cells are allowed full re-sensitization (this step is not clearly explained in the methods). For example, was fatty acid free BSA used for the migration medium? Were the red blood cells lysed efficiently and washed extensively with fatty acid free containing medium? Although it is estimated that the concentration of S1P in the MZ is lower than the blood, given that S1P3 may be involved, it could be useful to try higher concentrations of S1P to improve significance of results (up to 1 μ M).

5. In figure 4d treatment with S1P enhanced adhesion to ICAM-1 under static conditions. The relevance of this is not clear in the context of flow, especially given that under shear forces the authors describe increased motility (4c) on ICAM-1. Testing attachment under flow +/- S1P will help to address this point.

6. In figure 5, the authors use double labeling combined with histological analysis to assess positioning of WT and α PIX KO MZB cells. They find in the α PIX KO a single labeled population of MZB cells that is displaced to the outer layer of the marginal zone, a site they refer to as part of the red pulp. It is not clear why cells in this region, an area which has been shown to be exposed to flow and therefore to blood, are not labeled in 5min pulse. Importantly, the authors rely on histology-based analysis to make these arguments. This approach is extremely limiting and can be easily misleading.

The statistical significance of the data presented in this figure is not clear. The data in 'b' and 'c' are not quantified and the number of independent experiments and mice used in each experiment are not mentioned. It is not clear if littermate controls were used. In 'd' only 2 repeats were performed, again, not clear if this represents in vivo labeling of only 2 mice, or 2 experiments performed with an unknown number of animals. Furthermore, the gating strategy needs to be improved (e.g. in 'd' it appears that the KO cells have lower single for CD21, thus biasing the number of double positives calculated).

α PIX KO MZB cells are shown to be normally positioned in the follicles (figure 5d), suggesting

entry is intact. It is therefore difficult to explain the dramatic reduction in Ficoll deposition based on impaired MZB cell access to the follicles. Given that this is a full α PIX KO, other defects could account for this observation.

With these concerns in mind, it not possible to conclude anything about the positioning of that the α PIX KO MZB are differentially positioned compared to WT.

7. Similarly, analysis of cell positioning in figure 7 and 8 is not convincing.

Minor comments:

1. Consistently across the paper, it is not clear what 'n' means; for example, does n=3 means 3 independent experiments performed with one mouse, or one experiment performed with 3 animals? In some figures (e.g. Figure 5d and 7c), graphs show SEM for n=2. It is not clear how this is calculated.

2. Introduction: in the last sentence of the second paragraph, 'CXCL5' should be 'CXCR5'.

3. Introduction: add that MZB do not recirculate 'in rodents'.

4. The use of the word 'towards' while describing migration against the flow is not clear and can lead to some confusion. This is an issue that comes up in multiple places in the manual script. Perhaps using 'up/down the flow' will be clearer?

5. In figure 3 the authors show that α PIX KO MZB cells migrate faster and detach more frequency on ICAM-1 and VCAM-1. However they do not show changes in directionality. The relevance of this behavior is not clear in the context of migration up or down the flow.

6. Text describing figure 3 (end of first paragraph, page 7): last sentence should be modified to state that loss of adhesion was selective to α PIX KO MZB cells migrating on ICMA-1, but not VCAM-1.

7. Figure 4: some clarity issues with regard to reproducibility of α PIX KO vs. WT behavior on ICAM1 and VCAM1. In the absence of S1P, straightness and migration index of WT and KO migrating on VCAM1 or ICAM1 are the same. Similarly, velocity of WT and KO migrating on VCAM-1 is the same. This is at odds with data presented in figure 3.

8. Figure 4c: not clear what is the experimental difference in this assay compared with figure 4b. If not different, why not using the larger data set described in this section to calculate migration

parameters of cells on ICAM1 presented in 'b'? Also, in section 'c' (displacement on VCAM-1)- were the data calculated from the same experiments described in 'b'?

9. Figure 4d shows adhesion in the presence of anti-CD11a, but there is no mentioning of this result in the text. Relevance should be explained, or data removed.

10. Figure 5d: legends describe intracellular staining of α PIX protein, but data only shows in vivo labeling.

11. Figure 6- tracks of migrating S1P3 KO cells showed decreased straightness compared with WT, which was independent of S1P. Suggested potential S1P-independent contribution of S1P3 to migration by sensing shear forces. Mechanistically, not clear how. This point could potentially be improved if S1P residues are better removed as suggested above for figure 4.

12. Figure 7- legend should indicate the mice used are chimeras

Reviewer #3 (Remarks to the Author):

Tedford et al. propose a mechanism by which splenic marginal zone B cells (MZB) migrate with or against the blood flow between the splenic follicle, marginal zone and red pulp. Using mainly in vitro flow chamber experiments, they show that MZB migrate against the blood flow on ICAM-1, but not VCAM-1) and this upstream migration was inhibited by S1P-S1PR3 signaling and enhanced in α PIX deficient MZB. By contrast, migration with the flow was observed on VCAM-1 and enhanced by S1P-S1PR3 signaling. Based on these results, the authors propose an interesting concept to explain the unique intrasplenic distribution of MZB and the dynamic shuttling of captured antigen by MZB to splenic follicles. However, while the idea is intriguing, the very reductionist in vitro approach, without in vivo confirmation of MZB migration dynamics tempers enthusiasm. Specifically, the authors may consider the following:

Major comments

1. It is unclear whether the in vitro assays exploring MZB migration under flow on ICAM-1 or VCAM-1 and the effect of the S1P pathway can do justice to the in vivo situation. First, it is puzzling why no experiments were performed with MAdCAM-1, which (unlike VCAM-1) is well known to be expressed in the MZ sinus. In fact, there are several other relevant adhesion molecules co-expressed in the MZ, including e.g. ICAM-2 and CD31, which may potentially impact MZB migration dynamics or directionality. Second, it is unclear whether the density of immobilized adhesion molecules used here is physiologically relevant. What was the effect of varying ICAM-1 coating densities on migratory behavior of WT and the various mutant MZB

that were used in this study? Third, although shear flow and S1P can modulate lymphocyte adhesion to endothelial cells, it seems likely that MZB in vivo are simultaneously exposed to a variety of chemokines. Although the authors speculate extensively on the role of chemokines in the discussion, no attempt was made to actually address their function experimentally.

2. In vivo relevance remains conjecture without direct visualization of MZB migration in intact spleen. Others have reported that the MZ in mice is accessible for intravital microscopy of MZB. Inclusion of such experiments would greatly strengthen this paper.

3. The authors show that α PIX KO MZB display altered migration patterns, but these cells are insufficiently described and characterized to interpret the results. For example, does α PIX deficiency alter the expression levels or function of integrins on MZB? Do mutant MZB show altered responses to chemokines?

4. I'm having difficulty to follow the authors' argument that MZB migration against blood flow would promote MZB migration into follicles. Blood does not enter the MZ sinus from follicular parenchyma, but via central and transverse arterioles. Why would MZB not end up migrating into those feeding vessels?

Minor Comments

1. The authors should ensure that the axes and legends in their figures are appropriately labelled (x-axis on Fig. 1c, 2b, 3b,d, legend on Fig 7a, 8a,c).

2. The text claims repeatedly that blood flow in the MZ is turbulent, which seems rather unlikely. To my knowledge, true turbulent flow (based on Reynolds number) occurs only in large vessels, but has not been reported in microvasculature. Authors should either provide a compelling reference or refrain from using this term.

Reviewer #1 (Remarks to the Author):

The manuscript by Tedford et al adds to the mechanism on how Marginal Zone B cells shuttle in the spleen to carry antigen to the follicle to support an adaptive immune response. This movement has been studied previously by The Cyster lab that defined that the migratory pattern relied on GRK2 mediated S1PR1 desensitization and resensitization. Here this is investigated further to include migration against flow. I think the study is interesting but have a couple of concerns that I think should be addressed with experiments or in writing.

Major concerns:

- In the studies on VCAM and ICAM am lacking the control of not having any adhesion molecule there or an irrelevant protein.

We thank the reviewer for pointing out this oversight. We now include a figure (Supplementary Fig. 1) showing the effects of removing ICAM-1 and/or BSA from the flow migration assay. Without ICAM-1, no migration was possible: with a BSA blocking step, cells did not adhere to the slide; without a BSA blocking step, cells stuck firmly and did not migrate at all. Interestingly, with ICAM-1 but without the BSA blocking step, MZB could adhere and migrate under flow but their directional migration was severely hindered. We assume that there was too much non-specific adhesion in the absence of BSA that prevented them from migrating up the flow. Also, we tested another irrelevant protein besides BSA in response to Reviewer#3's suggestion to try PECAM-1 as a ligand: MZB could not adhere at all to the PECAM-1 and were completely washed out (data not shown). Additionally, we believe that the fact that we observed such a different type of migration on VCAM-1 as opposed to the ICAM-1 argues that the effects are specific to the ligands used.

- Which is the dominating force when comparing S1P1 and S1PR3 and how big is the expression difference? Also, the mechanism which by SiPR3 responds to flow should be further substantiated.

We attempted to quantify expression of S1PR1 and S1PR3 using flow cytometry analysis with antibodies against the 2 receptors but were not successful due to the failure of commercial antibodies. Because we have S1PR3 ko mice, we were able to test signals obtained from 3 different commercial antibodies on wildtype MZB cells using the ko mice as controls, and none of the antibodies gave a specific signal that was absent on the ko cells. However, the Immgen.org databank shows levels of S1PR1 RNA that are 4x higher than S1PR3 on MZB cells. On the other hand, S1PR1 is also expressed on numerous T cell populations, while S1PR3 appears to be specific to MZB cells.

At the suggestion of Reviewer #2, we have now removed Figure 7, which showed CYM effects on MZB from the four genotypes used in our study. Therefore we no longer address S1PR1 in our study. We believe that the study is improved by the focus on S1PR3, as we show a specific function for it in reversing the S1P-mediated inhibition of MZB migration up the flow (Figures 8 and 9).

With respect to the mechanism by which S1PR3 responds to flow in the absence of S1P, we do not have any further data to provide on this because it is likely that compensation by S1PR1 is masking the effects of S1PR3 loss, which were modest at best. S1PR1 can probably compensate

for S1PR3 absence because S1PR1 has an S1P-independent response to shear flow in endothelial cells (Jung B et al., Dev Cell, 2012). Givens and Tzima (Dev Cell, 2012) wrote an informative comment on the S1PR1 finding in which they compare it to 2 other receptors that have shown dual responses to either ligand, or absence of ligand plus mechanical stress. Given the 4-fold higher expression of S1PR1 over S1PR3 on MZB, even if S1PR1 is internalized faster than S1PR3, we believe it would only make sense to study this effect with both S1PR1 ko and S1PR3 ko mice or mice carrying only receptors with a mutated ligand-binding domain.

- Does the knocking out of α PIX alter the expression of receptors needed for transport of TNP-Ficoll by MZBs?

According to the suggestion of Reviewer #2, we have now removed the Ficoll experiment figure. However, we did test expression of CD21 on wildtype, α PIX ko, S1PR3 ko, and α PIX S1PR3 dko MZB cells, and found it normal (please see figure below).

Minor concerns:

-The flow of the spleen is often described a "turbulent current". However, the blood flow slows down significantly in the spleen due to the much larger volume when entering. Thus the overall description of the flow in the spleen could be discussed better.

We agree with the reviewer that the blood flow would slow down when it reaches the marginal zone and we have taken out the turbulent comments. Although the blood streams out of holes in the marginal sinus, and much like the jets in a hot tub, there would be more force where it emerges then farther away. However, we have no data on this and we restricted our descriptions to simply stating that there is blood flow from the direction of the follicle outwards towards the red pulp.

-This study would greatly be helped by adding a more complete mechanism picture to better explain the different forces affecting MZB migration.

We have now expanded our figure to show details more carefully and we hope that it will be helpful and more interesting (Fig. 10).

-What was the purity and viability of the sorted MZBs and FOBs?

We now include a statement that the purity of MZB cells sorted with the Miltenyi MZB cell purification kit named in the methods was over 90%. We also tested viability of purified MZB and FOB using the kit. Viability, as defined by lack of AnnexinV or 7AAD staining, was acceptable. We tested it immediately after purification when the cells were still cold, and we tested it after 30 minutes and 60 minutes of incubation at 37° and in all three conditions, viability was about 80% for both MZB and FOB cells (please see figure below).

a**b**
Supplementary Figure for Response to Reviewer #1:

a) Viability of purified MZB (left) and FOB cells (right). MZB cells were purified from spleen and stained for annexinV and 7AAD immediately and after 30' and 60' of incubation at 37°.

b) Expression of CD21 on α PIX ko, S1PR3 ko, and α PIX S1PR3 dko MZB cells.

Reviewer #2 (Remarks to the Author):

In the manuscript “The opposing forces of shear flow and sphingosine 1-phosphate control marginal zone B cell shuttling”, the authors use an elegant in vitro system to characterize the movement of MZB cells under shear forces. They show that ICAM-1 promoted migration of MZB cells towards the flow, while VCAM-1 enhanced adhesion and promoted movement in the opposite direction. Histological analysis showed that ICAM-1 was highly expressed in the marginal zone, while VCAM-1 was abundant in the red pulp. SIP signaling through SIP3 was suggested to counteract migration against flow, potentially leading to movement towards the red pulp. In addition, MZB cells derived from α PIX Rho-GEF KO mice showed increased motility under shear force, a defect that could be partially corrected when crossed with SIP3 KO mice. The authors also attempted to assess the distribution of MZB cells in mice in which α PIX Rho-GEF, SIP3 or LFA-1 function is impaired. They propose that MZB cell distribution is shifted towards the red pulp when α PIX Rho-GEF or LFA-1 are dysfunctional and that this phenotype depends largely on signaling via SIP3.

The model suggested by the in vitro data in this manual script is compelling, complements previous studies and generates new interesting hypotheses. Overall, the observations described in this work are novel and have the potential to stimulate new thoughts and exciting scientific discussion in the immunology and cell migration fields. However, we feel that in its current form, the manual script suffers from insufficient strength to support some of its main conclusions and can greatly benefit from an extensive revision. In addition, the in vivo data are not convincing and the authors should consider removing them altogether.

Major concerns:

1. The main claim of the paper is that MZB cells migrate in opposite directions in response to SIP and shear flow. This statement is largely based on quantification of cell velocity, track straightness and migration index under various conditions. However, direct quantification of directionality (specifically, movement up or down the flow) is missing. Such analysis will greatly improve the strength of the arguments.

The migration index parameter to quantify directionality is standard in the chemotaxis and flow migration fields (2 examples using T cells: Dominguez GA, Integrative Biol, 2015 or Valignat MP, Biophys J, 2013). To calculate the migration index, the end y-coordinate of a track is divided by the accumulated distance of the track, and these values are averaged for all the tracks. The flow in every movie and as depicted in every track plot comes from the top of the track plot in a line down the Y-axis. Thus if a cell migrates in a straight line up the y-axis, its directional migration value would be high, whereas one that meanders a while before ending at a point not much higher than the x-axis would have a low directional score. This is how we calculated it (using the Ibidi chemotaxis tool), but we also checked the results from dividing the counts of cells that ended above the x-axis with those that ended below the x-axis and found that the average of these ratios reflected the migration index. However, they were somewhat more erratic, therefore we stayed with migration index to measure directionality.

2. Figure 2, the authors claim that VCAM-1 expression is restricted to the red pulp and that in the absence of ICAM-1 it does not support migration against flow. These observations seem at odds with published works showing that expression of VCAM-1 alone is sufficient to allow retention in the marginal zone. Is it possible that VCAM-1 is expressed in lower levels in the marginal zone region and that its ability to enhance adhesion (as suggested by the authors) under flow is sufficient to promote retention in the absence of ICAM-1? Better histology may help to clarify this point. In addition, to further clarify the relationship between ICAM-1 and VCMA-1 in promoting directional migration of MZB cells, it will be useful to repeat the experiment described in figure 2a using a constant concentration of ICAM-1 mixed with increasing concentrations of VCAM-1 (an irrelevant protein can be added to maintain constant total protein density across the chamber's surface).

We agree that there must be a small amount of VCAM-1 in the marginal zone, but even with a higher exposure of the VCAM-1 stain, the amount of VCAM-1 signal was very low compared to ICAM-1 in the marginal zone or VCAM-1 outside the marginal zone. One paper on splenic macrophages used the absence of VCAM-1 staining to define the marginal zone boundary (Gonzalez NA et al., Nat Immunol; 2013). However, it is true that even if the amount of VCAM-1 is low it does have an effect on adhesion of the MZB cells, as shown by the experiments using blockade of VLA-4 and/or LFA-1 (Lu T et al., Science 2002; Song J et al., PNAS 2013).

To determine if a small amount of VCAM-1 affects flow-induced directional migration of MZB cells, we repeated the ICAM-1/VCAM-1 flow experiment using a range of amounts of VCAM-1 with a constant amount of ICAM-1, as suggested (Supplementary Figure 2). However, we did not observe any effects on migration until the ratio of ICAM-1 to VCAM-1 was almost 1:1. We also did not see a difference in detachment under flow (data not shown). However, we used a flow strength of only 3 dyn/cm², and it is possible the VCAM-1 in small amounts helps MZB to resist a stronger flow that could occur sporadically in the marginal zone.

3. Figure 3- the authors show that detachment of α PIX MZB cells from VCAM-1 under flow is similar to WT cells. Surprisingly, addition of ICAM-1 led to significantly increased detachment from VCAM-1. The physiological significance of these observations is not clear and it is somewhat surprising that reduced adhesion in vivo does not lead to gradual loss of the marginal zone compartment in α PIX KO. Comparing detachment of α PIX KO MZB and FOB cells may help to clarify this point.

The fact that the addition of ICAM-1 led to the significantly increased detachment of α PIX ko MZB from VCAM-1 could be explained by the fact that LFA-1 can inhibit VLA-4 binding to ligand (Uotila LM et al, JBC, 2014).

We agree with the point that the detachment data implies a gradual loss of α PIX ko MZB from the marginal zone compartment, because this is our hypothesis for the presence of α PIX ko MZB in the red pulp: that their reduced adhesion causes the flow to dislodge them over time. Although we would argue that the time frame over which the process occurs is probably longer than is implied by the detachment assay we show. For this detachment experiment, we used high levels of flow, up to 15 dyn/cm², to force detachment of MZB cells in the short time frame (30 minutes)

of imaging. We believe that it is unlikely that this level of shear force represents the in vivo situation in the marginal zone. If we were to measure detachment differences at 3 dyn/cm^2 , we would have to image for hours, which could be detrimental for cell viability. Thus the detachment differences between wildtype and $\alpha\text{PIX ko}$ MZB cells could induce gradual loss of $\alpha\text{PIX ko}$ MZB over a longer period of time, which would be more physiologically realistic and could explain the mislocalized MZB in $\alpha\text{PIX ko}$ spleens.

4. In figure 4 the authors claim that SIP promotes adhesion and motility but inhibits directional migration of MZB against the flow. However, the effect induced by SIP treatment on WT is not convincing and the absence of quantitative analysis of directionality (e.g. how many cells go up or down the flow?) makes it difficult to interpret the results.

It is possible that the relatively modest effect of SIP in these assays can be improved if cells are allowed full re-sensitization (this step is not clearly explained in the methods). For example, was fatty acid free BSA used for the migration medium? Were the red blood cells lysed efficiently and washed extensively with fatty acid free containing medium? Although it is estimated that the concentration of SIP in the MZ is lower than the blood, given that SIP3 may be involved, it could be useful to try higher concentrations of SIP to improve significance of results (up to $1 \mu\text{M}$).

We thank the reviewer for this suggestion to repeat the SIP experiment with increased amounts of SIP. We first started with a range from 200 nM to $1 \mu\text{M}$ but found little differences above 600 nM and so we tested 0, 200 nM and 600 nM and presented the results in the new Figure 5. In this experiment, there was a significant difference between untreated and 200 nM in inhibiting migration against flow. Significance for a modest effect is hard to achieve with the relatively small number of data points we use, each one representing the average of over 100 cells from an individual mouse, but we chose to not count individual tracks, even though that would have greatly increased statistical significance, in order to avoid the statistical problem of pseudoreplication. However, even a modest effect in a 30 minute imaging time frame can have large effects in vivo longer time spans. And although the effects of SIP on wildtype MZB are relatively small compared to the effects of SIP on $\alpha\text{PIX ko}$ MZB cells, they are clear and consistent.

We don't believe that residual SIP from the purification is present or affecting the MZB migration because RBCs were removed in 2 steps from the procedure: first, RBCs were lysed immediately after the spleen cell suspension was made and second, an anti-Ter119 antibody was present in the antibody cocktail provided by the Miltenyi kit that removes non-B cells from the cell suspension. The purification procedure, which takes about 4 hours, includes multiple washing steps with PBS containing fatty acid-free BSA. No other BSA or serum touched the cells. Additionally, MZB cells were incubated on the slides at 37° for 30 minutes prior to migration, which would have allowed the re-sensitisation of the SIP receptors. We now include this information in the Methods section. Also, we compared the results of the migration index calculations to the ratio of counts up to counts down and again: the two ratios reflect each other well, but the counts up to down ratio values vary a bit more so we used the migration index instead.

5. In figure 4d treatment with S1P enhanced adhesion to ICAM-1 under static conditions. The relevance of this is not clear in the context of flow, especially given that under shear forces the authors describe increased motility (4c) on ICAM-1. Testing attachment under flow +/- S1P will help to address this point.

We compared the detachment of MZB cells treated or not with S1P (new Fig. 5c) and found no differences. This is interesting because S1P inhibits migration under flow and increases static adhesion, but does not affect detachment under flow. We conclude from this that S1P affects placement of integrin adhesions, from polarized at the leading edge to peripheral locations around the cell. We found that S1P (200 nM) treatment increases the % of cells that migrate more than a few cell lengths from their starting point (10 μ m). If these weaker cells are unable to polarize integrin complexes in the direction of flow, then S1P may provide enough power to move by ensuring productive integrin complexes around the periphery of the cell.

6. In figure 5, the authors use double labeling combined with histological analysis to assess positioning of WT and α PIX KO MZB cells. They find in the α PIX KO a single labeled population of MZB cells that is displaced to the outer layer of the marginal zone, a site they refer to as part of the red pulp. It is not clear why cells in this region, an area which has been shown to be exposed to flow and therefore to blood, are not labeled in 5min pulse. Importantly, the authors rely on histology-based analysis to make these arguments. This approach is extremely limiting and can be easily misleading.

The statistical significance of the data presented in this figure is not clear. The data in 'b' and 'c' are not quantified and the number of independent experiments and mice used in each experiment are not mentioned. It is not clear if littermate controls were used. In 'd' only 2 repeats were performed, again, not clear if this represents in vivo labeling of only 2 mice, or 2 experiments performed with an unknown number of animals. Furthermore, the gating strategy needs to be improved (e.g. in 'd' it appears that the KO cells have lower single for CD21, thus biasing the number of double positives calculated).

α PIX KO MZB cells are shown to be normally positioned in the follicles (figure 5d), suggesting entry is intact. It is therefore difficult to explain the dramatic reduction in Ficoll deposition based on impaired MZB cell access to the follicles. Given that this is a full α PIX KO, other defects could account for this observation.

We agree and we have now removed the Ficoll figure, as our study is focused on what happens to α PIX ko MZB cells in the marginal zone and not inside the follicle.

With these concerns in mind, it not possible to conclude anything about the positioning of that the α PIX KO MZB are differentially positioned compared to WT.

7. Similarly, analysis of cell positioning in figure 7 and 8 is not convincing.

We are again thankful for these observations and have now completely removed the double in vivo labeling data. We now use a 5 minute α CD21-PE injection to delimit the borders of the marginal zone and co-stain the spleen sections with anti-IgM to reveal B cells positioned outside the boundary of the marginal zone. We define "red pulp" as the area beyond the cells stained with 1 μ g of α CD21 in a 5 minute injection. To validate the use of α CD21 as a marker of the marginal

zone, we titrated the amount of injected α CD21 (new Supplementary Figure 3) to show that high amounts (5 μ g) will mark all IgM⁺ cells in α PIX ko mice, however, at the comparatively lower level of 1 μ g that we use, the outsiders are not stained with CD21 but with IgM. This effect is not only specific to α PIX ko MZB cells, as an injection of α LFA-1 also dislodges MZB cells so that some appear just outside the CD21-marked boundary. It is unclear why this portion remains unstained by α CD21, but we speculate that it is because these cells are located in an area with high VCAM-1 levels. Thus the limiting amounts of CD21 are preferentially taken up by the MZB cells in the marginal zone that can move freely, while the MZB cells in the VCAM-1 area may be hindered from moving due to increased adhesion on VCAM-1. In addition, the CD21-labeling antibody might be efficiently bound by MZB cells closer to the sinus, leaving less antibody for staining cells at the border to the red pulp. Nevertheless, the method does reveal the MZB outside the marginal zone and can be used to show that loss of S1PR3 restores the localization of the α PIX ko MZB to the marginal zone. This data is supported by the flow migration data showing that loss of S1PR3 prevents S1P from inhibiting α PIX ko MZB migration up the flow (Fig. 8). To strengthen the histology data, we followed the helpful suggestion that we quantify the histology findings, and used ImageJ to measure mean gray intensity inside defined areas drawn around the CD21 border. We added the description of the quantification method to the Methods section.

We also created a new figure by moving the α LFA-1 injection data to an earlier position in the manuscript and paired it with a new experiment testing the effects of cutting off blood supply to the spleen to answer Reviewer #3's request for in vivo relevance. This figure is intended to show the effects of blood flow on MZB cell positioning.

Minor comments:

1. Consistently across the paper, it is not clear what 'n' means; for example, does n=3 means 3 independent experiments performed with one mouse, or one experiment performed with 3 animals? In some figures (e.g. Figure 5d and 7c), graphs show SEM for n=2. It is not clear how this is calculated.

We apologize for the lack of detail and have now included the necessary information in the figure legends. In the 2 instances cited, the graphs actually showed mean \pm SD and we mistakenly wrote SEM, but those 2 graphs have now been removed. We now changed most graphs to show mean \pm SEM except for 2 graphs that were left as bars showing mean overlaid with symbols.

2. Introduction: in the last sentence of the second paragraph, 'CXCL5' should be 'CXCR5'.

We fixed this mistake.

3. Introduction: add that MZB do not recirculate 'in rodents'.

We fixed this too.

4. The use of the word 'towards' while describing migration against the flow is not clear and can lead to some confusion. This is an issue that comes up in multiple places in the manual script. Perhaps using 'up/down the flow' will be clearer?

We agree, and we changed all instances of against to up/down the flow, and we believe that this does make the writing clearer.

5. In figure 3 the authors show that α PIX KO MZB cells migrate faster and detach more frequency on ICAM-1 and VCAM-1. However they do not show changes in directionality. The relevance of this behavior is not clear in the context of migration up or down the flow.

We argue that α PIX ko MZB cells can migrate directionally against the flow but have dynamic adhesion defect that results in flow pushing them out of the marginal zone while they are migrating. Although they do not detach more on VCAM-1, probably because it is so sticky, their loose adherence or increased migration speed means that they migrate down the flow on VCAM-1 more directly than wildtype MZB cells do. Both of these factors would increase their chances of moving out of the marginal zone and into the red pulp area. These data support the histology finding that α PIX ko MZB cells are mislocalized in the red pulp and provide evidence for our model that blood flow (along with S1P) is an important determinant of MZB cell positioning.

6. Text describing figure 3 (end of first paragraph, page 7): last sentence should be modified to state that loss of adhesion was selective to α PIX KO MZB cells migrating on ICMA-1, but not VCAM-1.

We made this change.

7. Figure 4: some clarity issues with regard to reproducibility of α PIX KO vs. WT behavior on ICAM1 and VCAM1. In the absence of S1P, straightness and migration index of WT and KO migrating on VCAM1 or ICAM1 are the same. Similarly, velocity of WT and KO migrating on VCAM-1 is the same. This is at odds with data presented in figure 3.

We apologize for this inconsistency. α PIX ko MZB cells normally show moderately higher values for straightness and migration index than wildtype on ICAM-1 and in this figure these values are the same as wildtype. It's not clear why this occurred, but it was one of the earliest experiments we performed. We now repeated the experiment, WT vs PIX migration on ICAM-1 with a new set of mice and have replaced this data set.

The increased velocity α PIX ko MZB cells compared to wildtype in the previous Figure 3 was on high concentrations of VCAM-1: 15 and 25 μ g/ml. We did this experiment to mimic what we thought VCAM-1 concentrations might be in the red pulp. In the following figure (old Figure 4, with S1P treatment of α PIX ko MZB), the VCAM-1 used is at a lower concentration (2.5 μ g/ml), which is probably why the velocity of wildtype and α PIX ko MZB cells are the same.

8. Figure 4c: not clear what is the experimental difference in this assay compared with figure 4b. If not different, why not using the larger data set described in this section to calculate migration parameters of cells on ICAM1 presented in 'b'? Also, in section 'c' (displacement on VCAM-1)- were the data calculated form the same experiments described in 'b'?

The larger set was because we rejected all of the directional migration data associated with these experiments since we were concerned that we did not have comparable cell density on the slides. If cell density differs, the cell-cell collisions could affect migration up the flow. However, the % increase of cells migrating more than 10 μm was not affected by varying cell densities. In any case, we removed this data set as explained in the answer to the preceding point. In the new data set, the difference with a smaller set of mice shows a significance of $p=0.09$.

9. Figure 4d shows adhesion in the presence of anti-CD11a, but there is no mentioning of this result in the text. Relevance should be explained, or data removed.

We were using it as a control to show that the static adhesion was LFA-1 dependent, but we removed it now.

10. Figure 5d: legends describe intracellular staining of αPIX protein, but data only shows in vivo labeling.

We removed this figure when we removed the double in vivo labeling to show movement of MZB into the red pulp.

11. Figure 6- tracks of migrating S1P3 KO cells showed decreased straightness compared with WT, which was independent of S1P. Suggested potential S1P-independent contribution of S1P3 to migration by sensing shear forces. Mechanistically, not clear how. This point could potentially be improved if S1P residues are better removed as suggested above for figure 4.

We would argue that S1P was efficiently removed, as described in our response to Major point 4, above. The slight difference in straightness of S1PR3 ko MZB cells could be explained by a ligand-independent function of S1PR3 in shear flow induced migration. With respect to the mechanism, we do not have any further data to provide on this because it is likely that compensation by S1PR1 is masking the effects of S1PR3 loss, which were modest at best. S1PR1 may compensate for S1PR3 absence because S1PR1 has an S1P-independent response to shear flow in endothelial cells (Jung B et al., Dev Cell, 2012). Givens and Tzima (Dev Cell, 2012) wrote an informative comment on the S1PR1 finding in which they compare it to 2 other receptors that have shown dual responses to either ligand, or to absence of ligand plus mechanical stress. Given the 4-fold higher expression of S1PR1 over S1PR3 on MZB, even if S1PR1 is internalized faster than S1PR3, we believe it would only make sense to study this effect with both S1PR1 ko and S1PR3 ko mice or mice carrying only receptors with a mutated ligand-binding domain.

12. Figure 7- legend should indicate the mice used are chimeras

We now state that the mice used in the new version of Figure 7 (now called Figure 9) were chimeras.

Reviewer #3 (Remarks to the Author):

Tedford et al. propose a mechanism by which splenic marginal zone B cells (MZB) migrate with or against the blood flow between the splenic follicle, marginal zone and red pulp. Using mainly in vitro flow chamber experiments, they show that MZB migrate against the blood flow on ICAM-1, but not VCAM-1) and this upstream migration was inhibited by SIP-SIPR3 signaling and enhanced in α PIX deficient MZB. By contrast, migration with the flow was observed on VCAM-1 and enhanced by SIP-SIPR3 signaling. Based on these results, the authors propose an interesting concept to explain the unique intrasplenic distribution of MAB and the dynamic shuttling of captured antigen by MZB to splenic follicles. However, while the idea is intriguing, the very reductionist in vitro approach, without in vivo confirmation of MZB migration dynamics tempers enthusiasm. Specifically, the authors may consider the following:

Major comments

1. It is unclear whether the in vitro assays exploring MZB migration under flow on ICAM-1 or VCAM-1 and the effect of the SIP pathway can do justice to the in vivo situation. First, it is puzzling why no experiments were performed with MAdCAM-1, which (unlike VCAM-1) is well known to be expressed in the MZ sinus. In fact, there are several other relevant adhesion molecules co-expressed in the MZ, including e.g. ICAM-2 and CD31, which may potentially impact MZB migration dynamics or directionality. Second, it is unclear whether the density of immobilized adhesion molecules used here is physiologically relevant. What was the effect of varying ICAM-1 coating densities on migratory behavior of WT and the various mutant MZB that were used in this study? Third, although shear flow and SIP can modulate lymphocyte adhesion to endothelial cells, it seems likely that MZB in vivo are simultaneously exposed to a variety of chemokines. Although the authors speculate extensively on the role of chemokines in the discussion, no attempt was made to actually address their function experimentally.

We thank the reviewer for these helpful suggestions. We repeated the MZB flow migration assays using MAdCAM-1, ICAM-2, PECAM-1 ligands and CXCL12 and CXCL13 chemokines (coated along with ICAM-1), and made some discoveries that became the basis for the new Figure 3 and part of our model. MZB did not adhere at all to CD31 (PECAM-1) and were completely washed out by the force of the flow (data not shown). Migration up the flow was normal on ICAM-2. However, MZB cell migration up the flow on MAdCAM-1 was completely inhibited. The cells were so tightly adherent that they could not move at all. We presume that this level of adhesion would allow the MZB cells to withstand any strong shear force close to the source, on the marginal sinus, and would also allow them to crawl through the sinus layer to get inside the follicle. Similarly, migration up flow on slides co-coated with ICAM-1 and CXCL13 was also inhibited, and CXCL13 expression is strong inside the follicle and around the sinus layer. These findings fill in a piece of the puzzle of how MZB cells migrate up the flow but do not keep going into the feeding vessels. We added this information to our new model slide (new Figure 10).

2. In vivo relevance remains conjecture without direct visualization of MZB migration in intact spleen. Others have reported that the MZ in mice is accessible for intravital microscopy of MZB. Inclusion of such experiments would greatly strengthen this paper.

We considered doing 2 photon microscopy for this manuscript, but had to defer the experiments for now due to the extremely challenging aspects of imaging the spleen in vivo. Although there has been one high profile paper using 2 PM on a spleen in situ (Arnon T et al., Nature, 2013), almost all other papers showing 2 PM of spleen are using explants – cutting the spleen out and studying it in a dish. This is because of the technical difficulties in imaging through the capsule, while the mouse is breathing and the spleen is hooked up to a blood supply which pulses, making it move. Ferrer et al. write: “The spleen is a particularly difficult organ for imaging as it accounts for a three-dimensional branched vasculature, containing both closed/rapid and open/slow circulations, and a compartmentalized parenchyma (red pulp, white pulp and marginal zone) enclosed within a dense capsule” (Ferrer M et al. Parasitology International, 2014). Although we tried to do the imaging, the challenges in measuring and interrupting flow would not be surmountable in the immediate time frame.

Therefore, we devised a different experiment to show the importance of blood flow to MZB cell migration (new Figure 4). We exploited the observation by the Cyster group that MZB cells shuttling into the follicle can be stained and followed using an α CD21 injection. Cyster’s group showed that at 5 minutes after injection, stained MZB cells are only outside the marginal sinus. But at 30 minutes after injection, the stained MZB cells are found inside the marginal sinus. We repeated these conditions and added a third: at 5 minutes after injection, the blood supply to the spleen was surgically ligated. The spleens were then isolated at 30 minutes after injection. In this condition, the shuttling of the MZB inside the follicle was almost completely blocked. A small amount still occurred, and we speculated that these cells may have been already adherent to the sinus or close by. We argue that this experiment provides evidence for our theory that blood flow is necessary for MZB migration to the follicle and shuttling.

3. The authors show that α PIX KO MZB display altered migration patterns, but these cells are insufficiently described and characterized to interpret the results. For example, does α PIX deficiency alter the expression levels or function of integrins on MZB? Do mutant MZB show altered responses to chemokines?

We modified the text to try to describe the α PIX role in migration without exceeding space limitations. We wrote that α Pix ko cells are faster and have increased actin polymerization (Korthals M et al., JI, 2014). The α PIX homolog, β PIX, is involved in turnover of focal adhesions (Kuo JC et al, Nat Cell Biol, 2011). Thus β PIX, which is expressed in MZB cells, could lead to integrin defects resulting in reduced adhesion, leading to faster migration. We tested α Pix ko MZB cell migration in transwells, and found increased migration to all conditions tested, including basal levels, or with ICAM-1 + S1P, or CXCL13. However, expression of integrins was normal on α Pix ko cells (new Supplementary Figure 4).

4. I'm having difficulty to follow the authors' argument that MZB migration against blood flow would promote MZB migration into follicles. Blood does not enter the MZ sinus from follicular parenchyma, but via central and transverse arterioles. Why would MZB not end up migrating into those feeding vessels?

The feeding vessels terminate in the sinus, which has an outer wall that is perforated to allow blood to flow out, a bit like a spherical shower head. Thus from one central arteriole come many small streams. When a MZB migrates up the flow and hits the sinus, which is coated with

MAdCAM-1, as our new experiment reveals, it will adhere very strongly and this could enable it to withstand the force of the flow coming from any nearby outlets.

Minor Comments

1. The authors should ensure that the axes and legends in their figures are appropriately labelled (x-axis on Fig. 1c, 2b, 3b,d, legend on Fig 7a, 8a,c).

We fixed this error.

2. The text claims repeatedly that blood flow in the MZ is turbulent, which seems rather unlikely. To my knowledge, true turbulent flow (based on Reynolds number) occurs only in large vessels, but has not been reported in microvasculature. Authors should either provide a compelling reference or refrain from using this term.

We have now removed the term and just refer more generally to blood flow.

Reviewers' comments:

Reviewer #1 (Remarks to the Author):

The authors have adequately addressed my concerns and I support publicaion.

Reviewer #2 (Remarks to the Author):

The authors have done a significant amount of revisions and I feel they adequately addressed many of the comments. However, major concerns remain, particularly with regards to the relevance or strength of some of the in vivo data.

1. The authors show that MZB cells can either adhere or migrate down the flow on VCAM-1. Which one is it? What is the basis for the difference in behaviour? Since VCAM-1 is known to work together with LFA-1 to prevent MZB displacement to the blood, it is not clear how promoting cell movement down the flow can achieve this. Can the authors clarify this and incorporate the conclusion in their model?

2. The authors refined their histology approach (described now in figure 4, 9 and suppl.2) and used it to reveal that a subtle change in MZB positioning occurs when LFA-1 is blocked.

While histology may indeed be the only way to assess such subtle changes in positioning, it is not a convincing approach to monitor the rate of MZB cells shuttling (figure 4b). If live imaging is too challenging, the authors should apply the more feasible and established quantitative approach using FACS-based analysis.

3. With these limitations in mind, the authors suggest that the rate of MZB entry to the FO is reduced when LFA-1 is blocked (figure 4b). If this is the case, mice treated with LFA-1 blocking antibodies for few days should show partial accumulation of MZB cells in the MZ. This should be tested in a quantitative manner using 5min single antibody i.v injection.

4. The authors suggest that in aPIX KO mice, MZB cells have migration defects that cause them to expand towards the 'red pulp' (figure 9), leading to partial displacement to the blood (suppl. Figure 6). If this was the case, I would expect that over time MZB cells will be lost not only from the MZ, but also from the spleen. This indeed can be a slow process and I do not suggest imaging times should be extended, but rather that a more careful assessment of the number of MZB cells in the KO mice is provided. Although in previous work reported by the same group MZB cell numbers were not reduced

(<http://mcb.asm.org/content/28/11/3776/F2.expansion.html>), it is possible that the effect would only be revealed in older mice. This should be tested and results discussed.

5. The aPIX KO cells are not well characterized and the nature of their defects remains confusing. The data in figure 6 suggests that aPIX KO cells have increased tendency to migrate both up (on ICAM) and down (on VCAM) the flow, why then in vivo the cells are displaced towards the 'red pulp'?

Instead, to establish a more direct link between opposing effects that are well supported by the in vitro system in this ms, the authors should test if LFA-1 blockade corrects the subtle shift of MZB to the 'red pulp' observed in the S1P3 chimeras.

As it is, it appears to me that the aPIX KO model does not provide support or additional clarification to the findings presented in the in vitro data and that, in fact, it only weakens the logic and consistency of the ms.

Minor comments:

1. The use of the term 'red pulp' to describe the outer layer of the MZ is confusing. Either the authors provide a more clear definition of the compartments in each of their histology figures, or rephrase the term.
2. Figure 2c- if the data is already published and is not adding to previous publication, should consider moving to suppl.
3. Figure 6a- should indicate the flow rate.
4. Page 11, second line from the top, $P = 0.9$ should be 0.09?
5. Page 11, figure 4d should be figure 7d?
6. Figure 4b, figure legends should clarify which antibody was given for 5/ 15min.
7. Page 12, line 3 “..there was a slight but significant decrease of straightness.. (figure 8a,b). P-value should be shown in the figure.
8. Reference to suppl. 5 is missing?
9. Just to clarify where is the typo; in Suppl. Figure 2, is the flow rate 8 dyn/cm² (as shown in the figure file), or 3 dyn/cm² (as indicated in the response to comments)?

Reviewer #3 (Remarks to the Author):

The authors have made a laudable effort and significantly improved their paper. Several of my comments have been well addressed, however, I still have two concerns:

1. The new in vivo experiments in Fig. 4 rely on transient interruption of blood flow to the spleen, which largely blocked the shuttling of MZB into follicles. While this result is consistent with the authors' hypothesis, there is a trivial (and more likely) alternative explanation. Several intravital imaging studies have reported that even brief periods of hypoxia lead to almost immediate arrest of migrating lymphocytes in lymphoid tissues. Therefore, the results in figure 4 do not provide rigorous in vivo support for this story.

2. The authors have not really provided an answer to my previous major comment #4, which stated: "Blood does not enter the MZ sinus from follicular parenchyma, but via central and transverse arterioles. Why would MZB not end up migrating into those feeding vessels?" Their response addresses how their study might explain MZB adhesion to MAdCAM in the marginal sinus, but they do not address why migration against shear flow would direct MZB into follicles rather than toward arterioles.

Reviewer #2 (Remarks to the Author):

1. The authors show that MZB cells can either adhere or migrate down the flow on VCAM-1. Which one is it? What is the basis for the difference in behaviour? Since VCAM-1 is known to work together with LFA-1 to prevent MZB displacement to the blood, it is not clear how promoting cell movement down the flow can achieve this. Can the authors clarify this and incorporate the conclusion in their model?

MZB cells can either adhere or migrate down the flow on VCAM-1, depending on flow strength and VCAM-1 concentration. On lower levels of VCAM-1, the MZB adhered enough to resist washout by shear flow but could only migrate minimally and not directionally up the flow. On higher levels of VCAM-1 and/or with high flow strength, the MZB moved down the flow as if being passively propelled by shear force. The α PIX ko MZB move even farther than wildtype down the flow, likely because they migrate faster than wildtype, displace longer distances, and are thus pushed farther by shear force. This MZB migration on VCAM-1 was in distinct contrast to how MZB migrate on ICAM-1, with many more cells migrating for considerably longer distances and in the direction of flow.

Our interpretation of this data is that VLA-4 and LFA-1 have very different functions on MZB, despite the fact that they are both involved in preventing MZB displacement to the blood. On T cells, LFA-1 and VLA-4 are both involved in migration but LFA-1 has some distinctive features. For example, in T cells migrating under shear flow, LFA-1 is distributed around the entire contact area while in contrast, VLA-4 is localized at the rear². Moreover, LFA-1 dominates over VLA-4 in supporting shear-resistant T cell crawling and trans-endothelial migration². Additionally, T cells migrating under shear flow move directionally up the flow when on ICAM-1 but move down the flow when on VCAM-1^{3,4}. One structural reason for these differences is that the alpha subunit of LFA-1 possesses an I-domain, an additional loop that the alpha subunit of VLA-4 lacks. The presence of the I-domain means that LFA-1 ligand binding is hindered relative to VLA-4 ligand binding and requires an additional activation signal¹. Thus for MZB, VLA-4/VCAM-1 binding may be easily activated by shear force, but does not support migration up the flow, while LFA-1 activation may be more complex since it must be organized so that the cell can polarize itself in the direction of shear force.

Collectively, our results suggest that VLA-4 may act as a backup adhesion system for MZB under flow. MZB migrating in the marginal zone are exposed to shear flow and high levels of ICAM-1, both of which induce MZB to migrate directionally up the flow. Directional migration on ICAM-1 implies that LFA-1 is distributed so as to position the cell along the axis of flow. However, if the flow direction changed, the MZB would have to re-position LFA-1. At this moment, the MZB would be vulnerable to being washed out by flow if it did not also have active VLA-4 to enhance adhesion. We do not mean to argue that VLA-4/VCAM-1 promotes cell movement down the flow because that implies an active process and we see this as a passive form of adhesion, not migration. In the absence of ICAM-1 and directional migration, MZB adhering to VCAM-1 are susceptible to the shear force effects of flow and either adhere in place under low flow or are pushed downstream by high flow. MZB are not normally found in the red pulp, however should an MZB reach the red pulp, its positioning would be determined by the proportions of ICAM-1 and VCAM-1 it encountered and the flow strength it was subject to. We do not know the in vivo relevance of the VCAM-1/ICAM-1 ratios in the red pulp, however,

VCAM-1 is expressed at very high levels on red pulp macrophages. These are often clustered around venous sinuses in the red pulp, thus it is possible that an additional VLA-4 function may be to ensure disposal of non-functional MZB through phagocytosis.

We thank the reviewers for this comment because it improved the manuscript. We apologize for not having addressed it sooner, and we have now included it in the manuscript in the introduction and discussion sections as paragraphs in bold font, and added it to the model Figure 9.

2. The authors refined their histology approach (described now in figure 4, 9 and suppl.2) and used it to reveal that a subtle change in MZB positioning occurs when LFA-1 is blocked.

While histology may indeed be the only way to assess such subtle changes in positioning, it is not a convincing approach to monitor the rate of MZB cells shuttling (figure 4b). If live imaging is too challenging, the authors should apply the more feasible and established quantitative approach using FACS-based analysis.

We began our studies of MZB positioning in spleen using FACS to identify MZB labeled in vivo with injected anti-CD21 and/or anti-CD35. The approach works well when MZB are positioned in the marginal zone or in the follicle because the staining results are binary: the MZB are either exposed to blood when in the marginal zone or not when in the follicle. However, when MZB are mislocalized to the red pulp, the FACS results are not so clear. The population of MZB that are CD35^{high} and CD21^{lo}, typically the shuttlers, could also be MZB in the red pulp. But MZB in the red pulp may be unevenly stained by antibodies in the blood due to variations in liquid diffusion through structural elements such as reticular meshwork and splenic cords, compounded by non-uniform draining through venous sinuses. We showed in Supplementary Fig. 3 that a titration of injected α CD21 amounts showed that MZB in the marginal zone and at the border with the red pulp were stained in proportion to the amount of α CD21, suggesting that the blood-containing antibody does not flow entirely freely through the compartments. Red pulp-localized MZB would thus be inconsistently labeled, resulting in misleading FACS analysis.

For these reasons, we used histology to view the reduction in shuttling MZB cells in anti-LFA-1-treated mice. While it is true that histology results can sometimes be unconvincing, we have now repeated this experiment 3 more times, each time with at least 3 mice, and have consistently seen a clear reduction in shuttling MZB cells inside the follicle (Fig. R1a). These results match those of the 1 hour LFA-1 blocking experiment that was already shown in Figure 4b. In summary, it may not be possible to use a FACS approach to quantify MZB cells when a subset of these is displaced to the red pulp. However, our histology shows highly consistent reductions in CD35⁺ shuttlers inside the follicle.

3. With these limitations in mind, the authors suggest that the rate of MZB entry to the FO is reduced when LFA-1 is blocked (figure 4b). If this is the case, mice treated with LFA-1 blocking antibodies for few days should show partial accumulation of MZB cells in the MZ. This should be tested in a quantitative manner using 5min single antibody i.v injection.

To attempt this experiment, we started with an injection of anti-LFA-1 for 24 hours, but when we saw the results it was clear that trying longer time periods would not be necessary. After 24 hours, the numbers of MZB cells in the spleen dropped by about 65% (Reviewer Fig. R1b). Using a 5 minute injection of anti-CD21 to label MZB in the marginal zone, we found by histology that the marginal zones were almost completely eroded in the spleens from 24 hour anti-LFA-1-treated mice (Fig. R1b). The relatively small amount of VCAM-1 in the marginal zone likely suffices to hold them there for a few hours but not for 24 hours with blocked LFA-1. Interestingly, the large amount of VCAM-1 in the red pulp was not enough to hold the bulk of the MZB in the absence of LFA-1. The remaining MZB cells appeared to be dispersed throughout the red pulp, showing that the loss of LFA-1 adhesion combined with blood flow through the marginal zone causes displacement of MZB cells down the flow and out to the red pulp. These results were surprising because marginal zones are present in $\beta 2^{-/-}$ ko mice ($\beta 2$ is a subunit of LFA-1)⁵. It is possible that the MZB in the $\beta 2^{-/-}$ ko mice are selected over time for the ability to compensate for the lack of LFA-1. In contrast, an injection of LFA-1 induces an acute loss of LFA-1, thus revealing the requirement for LFA-1 in correct MZB positioning.

We then compared the results of the 24 hour injection to those of a 2 hour injection. There was no drop in MZB numbers in the spleen at 2 hours (Fig. R1b), but histology of CD35+ MZB shuttlers inside the follicles showed a clear reduction that was consistent with the results of the 1 hour LFA-1-blocking injection presented in the manuscript in Figure 4b (Fig. R1a). We have now added Fig. R1b, the 24 hour α LFA-1 injection, to Figure 4 in the manuscript. The manuscript shows that LFA-1 supports MZB migration up flow and that there is substantially more ICAM-1 than VCAM-1 in the marginal zone. We argued above in the response to comment #1 that VLA-4 would provide an additional level of adhesion for the MZB, but these results and those of Lu et al show that it can only do this long enough to hold MZB for 3 hours but not 24 hours. Therefore, the results from these two LFA-1-blocking injection experiments provide support a dual role for LFA-1/ICAM-1 interactions: LFA-1 prevents MZB from being washed out of the marginal zone by the force of blood flow and supports migration up flow into the follicle.

4. The authors suggest that in α PIX KO mice, MZB cells have migration defects that cause them to expand towards the 'red pulp' (figure 9), leading to partial displacement to the blood (suppl. Figure 6). If this was the case, I would expect that over time MZB cells will be lost not only from the MZ, but also from the spleen. This indeed can be a slow process and I do not suggest imaging times should be extended, but rather that a more careful assessment of the number of MZB cells in the KO mice is provided. Although in previous work reported by the same group MZB cell numbers were not reduced (<http://mcb.asm.org/content/28/11/3776/F2.expansion.html>), it is possible that the effect would only be revealed in older mice. This should be tested and results discussed.

We assessed MZB cell numbers in older (20-40 weeks) α PIX ko mice, and compared them to the numbers in younger (8-10 weeks) mice. The reduced adhesion of α PIX ko MZB cells under flow would indeed suggest that their numbers would be reduced over time, but we found instead that α PIX ko MZB cell numbers were increased by about 50% in both younger and older mice (Fig. R2a/new Supplementary Fig. 4) and are also increased in chimeric α PIX ko and α PIX S1PR3 dko mice (Fig. R2b/new Supplementary Fig. 4). Despite the increased MZB numbers in the dko chimeric mice, MZB positioning in

the marginal zone was normal, indicating that increased MZB numbers do not cause mislocalization toward the red pulp.

These data point to a homeostasis mechanism that ensures the correct number of MZB cell numbers for the marginal zone. For example, iNKT control MZB numbers by inducing cell death and inhibiting MZB proliferation using a mechanism that requires direct contact between iNKT and MZB⁶. It has been shown that α PIX ko thymocytes fail to develop into T cells due to defective contacts with antigen-presenting cells that provide proliferation signals⁷. These failed contacts result from the increased velocity of α PIX ko thymocytes⁷. Thus it is possible that α PIX ko MZB are not able to make effective contacts with iNKT that would inhibit their proliferation and keep their numbers at a normal level. Consistent with this possibility, MZB cells carrying a combined mutation in α PIX and its binding partner GIT2 are faster than α PIX ko MZB cells, and are present in the spleen in even greater numbers than the α PIX ko MZB cells (our unpublished data). We have now included the data on α PIX ko MZB numbers in Supplementary Fig. 4.

5. The α PIX KO cells are not well characterized and the nature of their defects remains confusing. The data in figure 6 suggests that α PIX KO cells have increased tendency to migrate both up (on ICAM) and down (on VCAM) the flow, why then in vivo the cells are displaced towards the 'red pulp'?

Instead, to establish a more direct link between opposing effects that are well supported by the in vitro system in this ms, the authors should test if LFA-1 blockade corrects the subtle shift of MZB to the 'red pulp' observed in the S1P3 chimeras.

As it is, it appears to me that the α PIX KO model does not provide support or additional clarification to the findings presented in the in vitro data and that, in fact, it only weakens the logic and consistency of the ms.

We thank the reviewer for their thoughtful critique and we apologize for the lack of clarity regarding the α PIX ko MZB phenotype. To answer the first comment, that it is unclear how α PIX ko MZB are mislocalized beyond the marginal zone to the red pulp domain (Fig. 8a), we argue that the reduced dynamic adhesion that we observed in vitro (Fig. 6c) has more impact on MZB positioning than their increased migration speeds up the flow on ICAM-1 and down the flow on VCAM-1 (Fig. 6 and Suppl. Fig 6). In unpublished data from our lab on α PIX ko T cells, we found that α PIX represses cofilin activation of actin turnover: α PIX ko T cells have increased cofilin activation, actin turnover, and faster lamellipodia formation, all of which explain their increased speed. Increased cell speed is inversely correlated with adhesion⁸⁻¹⁰, suggesting that α PIX ko MZB have decreased adhesion. However, the α PIX ko MZB cells still express the homolog, β (beta)-PIX, meaning that integrin adhesion would only be reduced and not abolished. Decreased adhesion of α PIX ko MZB under flow, as we show in Fig. 6c, would lower their ability to withstand shear flow in the marginal zone and push them in the direction of the red pulp. Additionally, the blood flow that washes α PIX ko MZB out of the marginal zone contains S1P. We found that α PIX ko MZB respond strongly to S1P by almost ceasing to migrate up the flow (Fig. 6). We suspect that S1P inhibits MZB migration up flow by re-positioning LFA-1, which would be easier to do in α PIX ko MZB

due to their increased actin turnover and larger lamellipodia. The net effect of S1P and decreased adhesion on α PIX ko MZB leads to their mislocalization in the red pulp.

With respect to the second point, that we should show the anti-flow migration effects of S1P by using an LFA-blocking treatment instead of α PIX ko MZB cells, we believe that the results of blocking LFA-1 for 24 hours show that this is not possible (point #3 above). The anti-LFA-1 blocking antibodies produce an immediate and acute effect on MZB positioning by preventing their attachment to the marginal zone to an extent that is probably impossible for the S1PR3 mutation to overcome. In contrast, the α PIX ko mutation only partly reduces LFA-1-mediated dynamic adhesion – the α PIX ko MZB can still attach to ICAM-1 and use it to migrate up flow. Therefore the release from S1P effects conferred by the S1PR3 ko would be enough to allow α PIX ko MZB to migrate up the flow and become normally positioned in the marginal zone, as we showed in Figure 8. Furthermore, because S1P restrains MZB migration against flow on ICAM-1, it follows that S1P directly acts on LFA-1, likely by preventing it from polarizing in the direction of flow. With S1PR3 mutated, LFA-1 would be released from cytoskeletal constraints and be free to reposition according to flow, and thus blocking LFA-1 on these cells would lead to rapid detachment and washout of the MZB.

With the third point, the reviewers raise questions regarding the value of the α PIX ko model. We believe that the α PIX ko MZB model is ideal for illustrating the main conclusions of the paper – that MZB cell positioning is determined by flow and by S1P. However, our explanations lacked clarity, for which we apologize. To restate them, the α PIX ko MZB, with their reduced dynamic adhesion, can migrate up flow but are gradually washed out by it, thus the mislocalized MZB in the red pulp of the α PIX ko spleen reveal the direction of blood flow. Additionally, the effects of S1P on flow direction are more visible with α PIX ko MZB than with wildtype MZB, likely due to their decreased integrin function as shown in the detachment Fig. 6c. With wildtype MZB treated with S1P, it can be difficult to appreciate that subtle differences in migration direction over a 30 minute assay have strong in vivo effects. But the strong effect of S1P on α PIX ko MZB and the equally strong reversal in direction of α PIX ko MZB localization upon S1PR3 mutation underlines the role of S1P forcing MZB to resist the effects of flow.

To make a stronger case for α PIX ko MZB as an optimal system for studying flow and S1P, we have now re-organized the α PIX ko section of the manuscript by condensing the two α PIX ko figures (Figs. 6 and 7) into one. From the former Figure 6, the first parts a and b were redundant with Figure 7 and were removed, the second part, c - detachment under flow, was added to the former Figure 7, and the third part, d- migration on large amounts of VCAM-1, was moved to Supplementary Figure 6. We also added the preceding information to explain the mechanisms of the α PIX ko MZB in bold type paragraphs in the results and discussion sections. We hope that the more focused and concise α PIX ko MZB section improves the flow of the manuscript, and we thank the reviewers for strengthening this line of reasoning.

Minor comments:

1. The use of the term 'red pulp' to describe the outer layer of the MZ is confusing. Either the authors provide a more clear definition of the compartments in each of their histology figures, or rephrase the term.

We have now added an inset to Supplementary Fig. 3b to show a close-up of α PIX ko MZB cells interspersed with F4/80+ red pulp macrophages as support for our claim that the α PIX ko MZB cells end up in the red pulp. However we have also replaced the phrase “to the red pulp”, meaning “in the red pulp”, throughout the manuscript with “toward the red pulp”, meaning in the direction of the red pulp.

2. Figure 2c- if the data is already published and is not adding to previous publication, should consider moving to suppl.

Although there are histology images of splenic ICAM-1 and VCAM-1 published elsewhere in the literature, for our manuscript it is crucial to show them combined in one figure in an overlay in order to illustrate how the two compartments – marginal zone and red pulp – are delineated by the different expression patterns of the ligands. This explains how the different responses of MZB cells migrating on ICAM-1 or VCAM-1 result in their positioning in vivo. We think that the value in seeing an image in the main body of the manuscript more than offsets the space saved by moving it to Supplementary. We referenced a previous publication to make it clear that we are not the first researchers to stain for ICAM-1 and VCAM-1, however we were not able to find a good overlay image in the literature. Nonetheless, even if we were able to reference an overlay, it is much easier for readers if they have it right there on the figure.

3. Figure 6a- should indicate the flow rate.

We had the flow rate as a title – a single digit number with the units at the right end – over each track plot. We would have made it easier to read, except that we have now removed this data because it was redundant with the former Figure 7. We have now merged the former Figures 6 and 7 in a new Figure 6.

4. Page 11, second line from the top, P= 0.9 should be 0.09?

Yes, we have now fixed this typo.

5. Page 11, figure 4d should be figure 7d?

Yes, we have now re-numbered the figures correctly.

6. Figure 4b, figure legends should clarify which antibody was given for 5/ 15min.

We edited the sentence in the legends to read as follows:
 “MZB cells that shuttled into the follicle were identified by double in vivo labeling using first an i.v. injection of α CD35 for 15 minutes, followed by a second i.v. injection of α CD21 for 5 minutes.”

We also added “i.v.” next to the images and specified which was first and which was second.

7. Page 12, line 3 “..there was a slight but significant decrease of straightness.. (figure 8a,b). P-value should be shown in the figure.

It was shown: there is a bar going from wildtype to the S1P3 ko to indicate that two-way ANOVA found a “2 asterisk” significance, meaning $p < 0.01$, for a genotype difference

(actual p value was 0.0011). The difference in straightness is fairly small, so we called it “slight”, but nonetheless significant.

8. Reference to suppl. 5 is missing?

Yes, this was an oversight that we have now fixed. Supplementary Fig. 5 is now Supplementary Fig. 7.

9. Just to clarify where is the typo; in Suppl. Figure 2, is the flow rate 8 dyn/cm² (as shown in the figure file), or 3 dyn/cm² (as indicated in the response to comments)?

We apologize for the error: the flow rate was 8 dyn/cm² – the 3 dyn/cm² in the response was not correct. The figure legend and figure show the correct flow rate.

Reviewer #3 (Remarks to the Author):

The authors have made a laudable effort and significantly improved their paper. Several of my comments have been well addressed, however, I still have two concerns:

1. The new in vivo experiments in Fig. 4 rely on transient interruption of blood flow to the spleen, which largely blocked the shuttling of MZB into follicles. While this result is consistent with the authors' hypothesis, there is a trivial (and more likely) alternative explanation. Several intravital imaging studies have reported that even brief periods of hypoxia lead to almost immediate arrest of migrating lymphocytes in lymphoid tissues. Therefore, the results in figure 4 do not provide rigorous in vivo support for this story.

We agree with the reviewers that the spleen should have been oxygenated in the original clamping experiment. We have now repeated the experiment but with excised spleens in oxygenated media using a standard protocol from 2 photon imaging of explanted spleen¹¹ and the same pumps and equipment that we used in our lab for imaging explanted thymic lobes⁷. Briefly, spleens were excised at 5 minutes after the CD21-PE labeling injection and transferred to a dish in an incubator containing 37° medium that was bubbled continuously with 95% oxygen + 5% CO₂, using a peristaltic pump that was also placed in the incubator. Care was taken to ensure that the tubing in the dish did not create a stream that would move the spleen.

The results of the new experiment were entirely consistent with the previous ones that left the clamped spleen inside the body cavity. With a five minute injection of α CD21, the marginal zone was illuminated and no MZB cells were detected inside the follicle (Fig. R3a). With a 30 minute injection of α CD21, as expected from the results of Cinamon et al.¹², the MZB cells were found inside the follicle. However, if we clamped the spleen at 5 minutes following CD21 injection, rapidly excised it, and transferred it to a dish containing oxygenated medium for 25 more minutes (total 30 minutes), we observed an almost complete loss of MZB shuttling into the follicle, consistent with our previous result using clamped spleens that were left in situ and not oxygenated. We therefore replaced the previous spleen clamping figure with the new spleen excision data (n=3), and added 1 mouse each to the quantifications of the 5 minute control (for a total of n=5) and 30 minutes control (new total of n=5) (Figure R2). Additionally, the methods section was updated with the new technique.

2. The authors have not really provided an answer to my previous major comment #4, which stated: "Blood does not enter the MZ sinus from follicular parenchyma, but via central and transverse arterioles. Why would MZB not end up migrating into those feeding vessels?" Their response addresses how their study might explain MZB adhesion to MAdCAM in the marginal sinus, but they do not address why migration against shear flow would direct MZB into follicles rather than toward arterioles.

We apologize for not fully answering the question in our earlier submission. Our earlier, overly brief answer was that the sinus-lining MAdCAM-1⁺ vascular endothelial cells would be an unavoidable collision target for MZB cells entering the marginal sinus. We greatly appreciated the reviewers' earlier suggestion that we test MZB migration up flow on slides coated with CXCL13 or MAdCAM-1, as we found that MZB adhered to both but so strongly to MAdCAM that it prevented migration up flow (Fig. 3). From this, we concluded that MZB cells would not migrate up the capillaries/terminal arterioles that

supply blood to the marginal zone, as they would first have to pass a long gauntlet of MAdCAM-1+ cells, to which they would adhere and presumably from there, find their way into the follicles.

To expand on the earlier answer, we would describe the marginal sinus in more detail. The marginal sinus is the cavity formed by the anastomosed, or joined, ends of capillaries originating from the central arteriole. The marginal sinus envelops the follicle that forms around the central arteriole. MZB cells heading to the follicle from the marginal zone have to cross over the marginal sinus before entering the follicle. The sinus is perforated on the marginal zone side with openings large enough for cell passage in rat^{13,14} and mouse spleen¹⁵. However, the openings on the marginal zone side of the sinus are not necessarily directly across from the capillary openings on the follicular side (Fig. 3.11 in reference¹⁶). Therefore, an MZB cell entering the sinus through an opening would have to “turn right or left” to move along the sinus, parallel to the follicle, to reach a capillary. The marginal sinus appears to be about 3 lymphocyte-widths from side-to-side (Fig.4 in reference¹⁷; Fig. 3.11 in reference¹⁶), although they may appear smaller in histology images than they really are due to shrinkage during fixation. Nevertheless, the chances that an MZB could migrate any appreciable distance through the sinus and arrive at the opening of a capillary without hitting the sinus walls would seem low. MAdCAM-1 is expressed on the follicular side of the sinus along with CXCL13¹⁸, and both of these ligands would induce strong adherence of the MZB according to our data in Figure 3, followed by transmigration into the follicle.

On the other hand, if the MZB cells were able to avoid the MAdCAM-expressing cells on the follicular side of the sinus and make their way to a capillary, they might indeed migrate up the flow of the capillary. However, the same collision problem would occur: if they bumped into the walls of the capillary, they would likely encounter a MAdCAM-expressing cell and would then adhere. We were not able to find conclusive proof that the capillaries are lined with MAdCAM+1 cells, however, EE Schmidt wrote that the “white pulp capillaries terminate in the MS, their endothelial walls continuous with the endothelial cells lining the MS.”¹⁵ In addition, Zindl et al. looked for markers of marginal sinus cells that would co-stain with MAdCAM-1 and found ephrinB2 and Flk-1, and showed that these markers stain the marginal sinus, along with MAdCAM-1, but also the sinus connecting vessels, the branching arterioles (capillaries), and the central arteriole¹⁹. Thus these locations are likely also positive for MAdCAM-1 and would cause MZB cell arrest. Finally, CXCL13 acts as an arrest chemokine for B cells (not MZB cells) at high endothelial venules in lymph nodes, and enhances their adhesion to MAdCAM1²⁰. Therefore it is possible that CXCL13 in the vicinity of cells lining the marginal sinus and capillaries enhances adhesion of MZB cells to MAdCAM, further ensuring their entry into the follicle. In summary, the marginal sinus-lining cells express MAdCAM, to which MZB are strongly adherent, and are continuous with the branching arterioles (capillaries), so they likely promote MZB binding and inhibit prolonged migration up the flow. We added a sentence describing this in the first paragraph of the manuscript discussion section.

References

- 1 Chigaev, A. & Sklar, L. A. Aspects of VLA-4 and LFA-1 regulation that may contribute to rolling and firm adhesion. *Front Immunol* **3**, 242, doi:10.3389/fimmu.2012.00242 (2012).
- 2 Shulman, Z. *et al.* Lymphocyte crawling and transendothelial migration require chemokine triggering of high-affinity LFA-1 integrin. *Immunity* **30**, 384-396, doi:10.1016/j.immuni.2008.12.020 (2009).
- 3 Dominguez, G. A., Anderson, N. R. & Hammer, D. A. The direction of migration of T-lymphocytes under flow depends upon which adhesion receptors are engaged. *Integr Biol (Camb)* **7**, 345-355, doi:10.1039/c4ib00201f (2015).
- 4 Steiner, O. *et al.* Differential roles for endothelial ICAM-1, ICAM-2, and VCAM-1 in shear-resistant T cell arrest, polarization, and directed crawling on blood-brain barrier endothelium. *Journal of immunology (Baltimore, Md. : 1950)* **185**, 4846-4855, doi:10.4049/jimmunol.0903732 (2010).
- 5 Lu, T. T. & Cyster, J. G. Integrin-mediated long-term B cell retention in the splenic marginal zone. *Science* **297**, 409-412, doi:10.1126/science.1071632 (2002).
- 6 Wen, X., Yang, J. Q., Kim, P. J. & Singh, R. R. Homeostatic regulation of marginal zone B cells by invariant natural killer T cells. *PloS one* **6**, e26536, doi:10.1371/journal.pone.0026536 (2011).
- 7 Korthals, M. *et al.* alphaPIX RhoGEF supports positive selection by restraining migration and promoting arrest of thymocytes. *Journal of immunology (Baltimore, Md. : 1950)* **192**, 3228-3238, doi:10.4049/jimmunol.1302585 (2014).
- 8 Nordenfelt, P., Elliott, H. L. & Springer, T. A. Coordinated integrin activation by actin-dependent force during T-cell migration. *Nature communications* **7**, 13119, doi:10.1038/ncomms13119 (2016).
- 9 Sixt, M. & Raz, E. Editorial overview: Cell adhesion and migration. *Curr Opin Cell Biol* **36**, iv-vi, doi:10.1016/j.ceb.2015.09.004 (2015).
- 10 Valignat, M. P., Theodoly, O., Gucciardi, A., Hogg, N. & Lellouch, A. C. T lymphocytes orient against the direction of fluid flow during LFA-1-mediated migration. *Biophys J* **104**, 322-331, doi:10.1016/j.bpj.2012.12.007 (2013).
- 11 Barral, P., Sanchez-Nino, M. D., van Rooijen, N., Cerundolo, V. & Batista, F. D. The location of splenic NKT cells favours their rapid activation by blood-borne antigen. *The EMBO journal* **31**, 2378-2390, doi:10.1038/emboj.2012.87 (2012).
- 12 Cinamon, G., Zachariah, M. A., Lam, O. M., Foss, F. W., Jr. & Cyster, J. G. Follicular shuttling of marginal zone B cells facilitates antigen transport. *Nat Immunol* **9**, 54-62, doi:10.1038/ni1542 (2008).

- 13 Sasou, S., Madarame, T. & Satodate, R. Views of the endothelial surface of the marginal sinus in rat spleens using the scanning electron microscope. *Virchows Archiv. B, Cell pathology including molecular pathology* **40**, 117-120 (1982).
- 14 Sasou, S., Satodate, R. & Katsura, S. The marginal sinus in the perifollicular region of the rat spleen. *Cell and tissue research* **172**, 195-203 (1976).
- 15 Schmidt, E. E., MacDonald, I. C. & Groom, A. C. Comparative aspects of splenic microcirculatory pathways in mammals: the region bordering the white pulp. *Scanning microscopy* **7**, 613-628 (1993).
- 16 Bowdler, A. (Humana Press, New York, 2002).
- 17 Cesta, M. F. Normal structure, function, and histology of the spleen. *Toxicologic pathology* **34**, 455-465, doi:10.1080/01926230600867743 (2006).
- 18 Katakai, T. *et al.* Organizer-like reticular stromal cell layer common to adult secondary lymphoid organs. *Journal of immunology (Baltimore, Md. : 1950)* **181**, 6189-6200 (2008).
- 19 Zindl, C. L. *et al.* The lymphotoxin LTalpha(1)beta(2) controls postnatal and adult spleen marginal sinus vascular structure and function. *Immunity* **30**, 408-420, doi:10.1016/j.immuni.2009.01.010 (2009).
- 20 Kanemitsu, N. *et al.* CXCL13 is an arrest chemokine for B cells in high endothelial venules. *Blood* **106**, 2613-2618, doi:10.1182/blood-2005-01-0133 (2005).

Figure R1

Figure R2

a

b

Figure R3

Reviewer Figure Legends

Figure R1. LFA-1 blocking injections for 2 hours reduces shuttlers and for 24 hours strongly reduces MZB numbers.

(a) Immunofluorescence microscopy of CD35⁺ MZB cells shuttling into follicles following 2 hours of i.v. α LFA-1 injection. MZB cells that shuttled into the follicle were identified by double in vivo labeling using i.v. injection of α CD35 for 25 minutes, followed by i.v. injection of α CD21 for 5 minutes. Right panel: quantification of mean gray intensity in a 70 μ m band inside of the marginal sinus, determined by co-staining with MAdCAM-1 and CD169 (not shown). Data are representative of 2 additional independent experiments with at least 3 wildtype mice per condition. Symbols in one condition group denote individual follicles, at least 4 per mouse in one experiment. Arrows = CD35⁺ CD21⁻ shuttling MZB inside the follicle. Data are expressed as the mean \pm SEM. ** $p < 0.01$ by t -test. Scale bar, 100 μ m.

(b) Immunofluorescence microscopy of CD21⁺ MZB cells in the spleen following 24 hours of i.v. α LFA-1 injection. White arrows show thinned marginal zones and increased CD21 staining outside the marginal zone. Images representative of 4 experiments for control (n= 12) mice; 2 experiments each for α LFA-1 for 2 or 24 hours (n=7). Data are expressed as the mean \pm SEM. ** $p < 0.01$, **** $p < 0.0001$, by one-way ANOVA. Scale bar, 100 μ m.

Figure R2: Increased numbers of α PIX ko in spleens.

(a) Splenocytes were analyzed by flow cytometry and gated on IgM, CD1d, CD21 and CD23. Results are from 1 experiment for wildtype young mice (8-10 weeks) (n= 4) and α PIX ko young mice (n=3) and 1 experiment for old (20-40 weeks) mice (n= 5 mice each). Bars show mean \pm SEM. ** $p < 0.01$ by t -test.

(b) Flow cytometry quantification of Ly5.2⁺ MZB numbers in Ly5.1⁺ chimeric mice (n=6-7 mice per genotype). Bars show mean \pm SEM. **** $p < 0.001$ by one-way ANOVA.

Figure R3: Comparison of spleen clamping and spleen excision confirms role of blood flow in MZB shuttling.

(a) Confocal immunofluorescent images of follicles at 5 minutes after α CD21 injection (n=4 from earlier experiments plus n=1 from new experiment), 30 minutes after injection (n=4 from earlier experiments plus n=1 from new experiment), 30 minutes after injection with spleen clamped at 5 minutes (n=4 from earlier experiments), or 30 minutes after injection with the spleen excised at 5 minutes (n=3 from new experiments). Shuttling B cells were identified by CD21 staining inside the follicle whose boundaries were determined by MAdCAM-1 staining of the marginal sinus (shown as a white line), indicated by arrow. Scale bar, 50 μ m. Arrows = CD35⁺ CD21⁻ shuttling MZB inside the follicle.

(b) Quantification of CD21⁺ shuttling B cells. CD21 mean fluorescence intensity was quantified in ImageJ using defined bands of 35 μ m inside the follicle. Data represent Symbols in one condition group denote individual follicles, 4-5 per mouse. Data are expressed as the mean \pm SEM. **** $p < 0.0001$ by one-way ANOVA relative to middle column. Scale bar, 100 μ m.

Reviewer #2 (Remarks to the Author):

The authors have addressed the comments and I find the ms is clear and convincing at its current form. I therefore support publication.